# Forecasting Vegetation Condition with a Bayesian Auto-regressive Distributed Lags (BARDL) Model

Edward E. Salakpi[1], Peter D. Hurley[1,2], James M. Muthoka[3], Adam B. Barrett[4], Andrew Bowell[1,2], Seb Oliver[1,2], and Pedram Rowhani[3]

[1]The Data Intensive Science Centre, Department of Physics and Astronomy, University of Sussex, Brighton BN1 9QH, UK
[2]Astronomy Centre, Department of Physics and Astronomy, University of Sussex, Brighton BN1 9QH, UK
[3]Department of Geography, School of Global Studies, University of Sussex, Brighton BN1 9QJ, UK
[4]Sackler Centre for Consciousness Science, Department of Informatics, University of Sussex, Brighton BN1 9QJ, UK

**Correspondence:** Edward E. Salakpi (e.salakpi@sussex.ac.uk)

**Abstract.** Droughts form a large part of climate/weather-related disasters reported globally. In Africa, pastoralists living in the Arid and Semi-Arid Lands (ASALs) are the worse affected. Prolonged dry spells that cause vegetation stress in these regions have resulted in the loss of income and livelihoods. To curb this, global initiatives like the Paris Agreement and the United Nations recognised the need to establish Early Warning Systems (EWS) to save lives and livelihoods. Existing EWS use a combination of Satellite Earth Observation (EO) based biophysical indicators like the Vegetation Condition Index (VCI) and socio-economic factors to measure and monitor droughts. Most of these EWS rely on expert knowledge in estimating upcoming drought conditions without using forecast models. Recent research has shown that the use of robust algorithms like Auto-Regression, Gaussian Processes and Artificial Neural Networks can provide very skilled models for forecasting vegetation condition at short to medium range lead times. However, to enable preparedness for early action, forecasts with a longer lead time are needed. In a previous paper, a Gaussian Process model and an Auto-Regression model were used to forecast VCI in pastoral communities in Kenya. The objective of this research was to build on this work by developing an improved model that forecasts vegetation conditions at longer lead times. The premise of the this research was that vegetation condition is controlled by factors like precipitation and soil moisture thus, we used a Bayesian Auto-Regressive Distributed Lag (BARDL) modelling approach which enabled us to include the effects of lagged information from precipitation and soil moisture to improve VCI forecast. The results showed a $\sim$ 2-week gain in the forecast range compared to the univariate Auto-Regression model used as a baseline. The $R^2$ scores for the Bayesian ARDL model were 0.94, 0.85 and 0.74, compared to the Auto-Regression model's $R^2$ of 0.88, 0.77 and 0.65 for 6, 8 and 10 weeks lead time respectively.

## 1 Introduction

Drought events are amongst the most prevalent natural disasters reported globally and affect some 55 million people annually (Deleersnyder, 2018). In Africa, the devastating effects of droughts are mostly seen in the Arid and Semi-Arid Lands (ASALs), where people's lives and livelihoods mostly depend on agro-pastoral activities (Gebremeskel et al., 2019). Pastoralism in these regions contributes immensely to food security and local economies (Vatter, 2019). However, the ASALs grass- and shrublands,

which serve as the main source of fodder for the livestock are among the first to be hit by low rains and extreme temperature (FAO, 2018). These dry spells, when prolonged, adversely impact the food markets, income, and eventually leads to the loss of livelihoods (FAO, 2018). As a consequence, several drought early warning systems (EWS) have been developed to avert and minimise the impacts of these hazards.

Global initiatives, such as the 2015 Paris Agreement and the United Nation's Sustainable Development Goals (SDGs) recognise the importance of establishing robust EWS to save lives and livelihoods (UNFCCC, 2015). Existing EWS combine data on biophysical indicators that measure hazard risk with a series of socio-economic factors to account for vulnerability and exposure for early action. Satellite Earth Observation (EO) rainfall estimates and vegetation health are some of the datasets commonly used to monitor these drought conditions. The USAID's [1] Famine Early Warning Systems Network (FEWS NET) utilises household livelihood information, rainfall estimates and the Normalized Difference Vegetation Index (NDVI) to monitor drought and its impact on food security (FEWSNET, 2019). In Kenya, the National Drought Management Authority (NDMA) monitors EO based biophysical indicators in combination with forage, livestock conditions and socio-economic data to monitor and anticipate future drought scenarios for early finance and early action (Klisch and Atzberger, 2016; FAO, 2017).

Recent research has highlighted robust methods for forecasting biophysical indicators used to measure vegetation condition. AghaKouchak (2014) harnessed the persistence property in soil moisture with the ensemble streamflow prediction (ESP) to provide skilful forecasts of the standardized soil moisture index for up to two months ahead. Barrett et al. (2020) forecasted the Vegetation Condition Index (VCI) with Auto-Regression (AR) and Gaussian Process (GP) models using historical values of the same indicator. Both models performed well for lead times up to 6 weeks. Adede et al. (2019) used a multivariate approach that considered the effects of exogenous variables on VCI. The model was based on an Artificial Neural Network (ANN) and provided precise forecasts for one month lead time. Other related research studies involved the use of an Autoregressive Integrated Moving Average (ARIMA) and the Seasonal Auto-Regressive Integrated Moving Average (SARIMA) models to simulate and forecast Vegetation Temperature Condition Index (VTCI) (Han et al., 2010; Tian et al., 2016). Jalili et al. (2014) also used a Multilayer Perceptron (MLP) model and a Support Vector Machine (SVM) model to forecast Standard Precipitation Index using Normalised Difference Vegetation Index (NDVI), Temperature Condition Index (TCI) and VCI and input variables. The Gradient Boost Machine (GBM) model was also used by Nay et al. (2018) to predict vegetation health using the Enhanced Vegetation Index (EVI) as indicator. While these all these models gave good forecast accuracies for shortrange forecasts, forecasts with longer lead times beyond six weeks will provide disaster risk managers ample time to prepare and implement relief measures. Apart from the various models used in the research studies cited earlier, a number of different indicators were also used, however for this paper we used VCI mainly because it is the indicator used by our major stakeholder the NDMA. Secondly, the complex nature of agricultural droughts requires an indicator that adequately responds to changes in hydro-climatic and biophysical factors like rainfall, temperature and soil moisture level (Vicente-Serrano et al., 2012; Yihdego et al., 2019), which are amongst the properties of the NDVI used to derive VCI for this paper. Although based NDVI VCI has been extensively used in drought research, a comparative analysis by Bowell et al. (2021) showed that in ASAL regions

---

[1]United States Agency for International Development (USAID)

with sparse vegetation cover, VCI based on the Soil Adjusted Vegetation Index (SAVI) is a more suitable indicator due to the correction of the background effect from soil reflectance.

This paper aims to build on existing forecast initiatives and develop models that accurately forecast VCI at longer lead times. More specifically, our approach will include the interaction between the lagged information from indicators and variables like precipitation, soil moisture, and vegetation condition in an Auto-regressive distributed lag (ARDL) model (Gujarati, 2003; Pesaran and Shin, 1999). ARDL models are useful in situations where variable $Y_t$ at a time $t$ is influenced by other variables $X_t$ at time $t$ and the same variables at previous time steps $X_{t-i}$.

Parameter estimation with ARDL models has traditionally been carried out with a maximum likelihood approach which produces point estimates and often results in over-fitting leading to imprecise predictions (Martin, 2018). To address this, the ARDL model used in this work was implemented within a Bayesian framework which allows the incorporation of prior knowledge of the model parameters. This approach generates a posterior probability distribution for the model parameters which enables more accurate quantification of prediction uncertainties and allows for more robust risk analysis (Lambert, 2018).

## 2 Study Area and Data

### 2.1 Study Area

This research was conducted in 20 counties within the ASAL regions (figure 2) of Kenya where the predominant activities are pastoralism and wildlife conservation. The ASAL regions make up about 80% (46,000 km$^2$) of Kenya's total land area (Marigi et al., 2016), farmers in these regions rely heavily on pastures and grasslands as the main source of feed for their animals (Sibanda et al., 2017). However, the erratic weather patterns in the eastern African region makes Kenya prone to frequent drought events which poses a threat to the country's food security and economy as a whole (Gebremeskel et al., 2019). During the 2008-2011 droughts the Kenyan economy lost a total of 21.1 billion USD (Cabot Venton et al., 2012; Cenacchi, 2014). Hence the need to develop a drought EWS with the ability to provide timely warnings for drought preparedness.

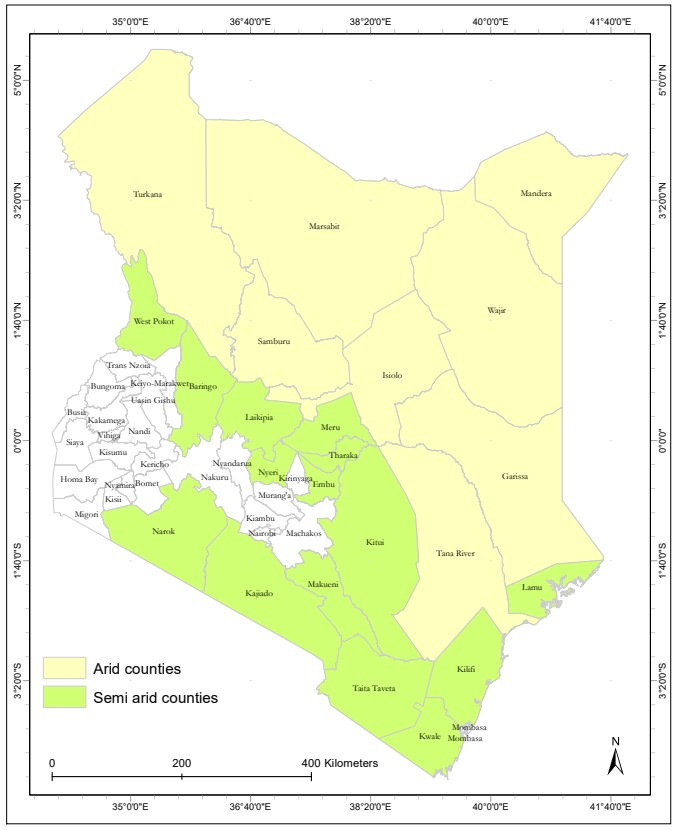

**Figure 1.** A map of Kenya showing the arid and semi-arid counties where the research was focused. Data was sampled from 8 arid and 12 semi-arid counties

## 2.2 Data

Developing a highly skilled model required adequate historical data on drought indicators and biophysical factors acquired
over a long period. Table 1 shows details of the satellite earth observation data used for this work.

**Table 1.** Summary of the datasets for the forecast model

| Data | Source (Producer) | Spatial Resolution | Temporal Resolution | Acquisition Period | Unit of Measure |
|------|-------------------|--------------------|--------------------|--------------------|-----------------|
| Precipitation | Climate Hazards Group InfraRed Precipitation (CHIRPS) | 5km | Daily | 2001-2018 | mm |
| Soil Moisture | European Space Agency's Climate Change Initiative (CCI) | 30km | Daily | 2001-2018 | $m^3/m^{-3}$ |
| Surface Reflectances | NASA MODIS (MCD43A4 v006) | 500m | Daily | 2001-2018 | N/A |

### 2.2.1 Precipitation (Rainfall Estimates)

The precipitation data were acquired from the Climate Hazards Group InfraRed Precipitation (CHIRPS) project (Funk et al., 2015). The CHIRPS data comprise a combination of weather station data and rainfall estimates captured via satellite remote sensing using the Cold Cloud Duration (CCD) (Milford and Dugdale, 1990) approach. The approach is used to estimate rainfall by using remotely sensed information on the period of time a cloud remains at a given temperature. The dataset is available as daily 5km resolution images.

### 2.2.2 Soil Moisture

The daily 30km resolution soil moisture products by the European Space Agency's Climate Change Initiative (ESA-CCI) was used for this work. The data is produced from an algorithm that takes in back-scatter information from multiple active and passive Synthetic Aperture Radar (SAR) satellites. The values generated represent soil moisture at a soil depth of 10cm. The ESA-CCI Soil moisture products are available as passive, active or a combination of both. For this work, the combined version of the data is used (Gruber et al., 2019; Dorigo et al., 2017; Yang et al., 2017).

### 2.2.3 Surface Reflectance

The bidirectional reflectance distribution function (BRDF) corrected MODIS product, MCD43A4 Version 6, (Schaaf and Wang, 2015) was used to compute the NDVI and VCI. The product is available as daily 500m resolution images captured in 7 bands ranging from visible to infrared. Information on the vegetation health is derived from the Red and Near-Infra Red(NIR) bands via equation (1).

$$\text{NDVI} = \frac{NIR - Red}{NIR + Red} \qquad (1)$$

## 3 Methods

### 3.1 Data pre-processing

The datasets were acquired from January 1, 2001, to December 31, 2018, to correspond with the availability of soil moisture data at the time of research. Apart from the precipitation, clouded and low-quality pixels from poor atmospheric and radiometric correction were removed using the quality flags from the Quality Assurance (QA) maps that came with the surface reflectance and soil moisture products. Pixels representing grasslands and shrublands areas within our regions of interest were retrieved with the European Space Agency (ESA)'s 2016 Sentinel 2 Land Use and Land Cover (LULC) map [2] Ramoino et al. (2018). For the coastal semi-arid counties like Lamu and Kwale we could not extract enough soil moisture data so no results were shown for these counties.

To measure the drought condition at a period in time, the minimum and maximum NDVI values for a chosen baseline time interval and the NDVI value for that period are used to compute the Vegetation Condition Index (VCI) via equation (2) (Kogan, 1995). VCI values range from 0-100, with values below 35 depicting a moderate to severe drought condition (Klisch and Atzberger, 2016).

$$\text{VCI}_i = 100 \times \frac{\text{NDVI}_i - \text{NDVI}_{\min,i}}{NDVI_{\max,i} - NDVI_{\min,i}}, \tag{2}$$

where $\text{VCI}_i$ is the VCI value to be derived for the $i^{th}$ week, the $\text{NDVI}_i$ is the NDVI values for $i^{th}$ week, $NDVI_{\min,i}$ and $NDVI_{\max,i}$ are the long-term minimum and long-term maximum NDVI values of a pixel at $i^{th}$ week of the year.

The percentage of temporal gaps created by the removal of clouded and poor quality pixels varied per county and ranged from as low as 0.1% to 35%. Counties with over 50% missing data were dropped. Gaps were were filled with the Radial Basis Function (RBF) interpolation method. This approach uses weighted basis functions derived from Euclidean distances to approximate missing values (Rippa, 1999). The advantage of using the RBF method was to ensure approximated values did not fall outside the valid ranges, especially over periods with long gaps. Noise resulting from faulty instruments were reduced with the Whittaker smoother (Eilers, 2003), which filters noise via a penalised least-squares. The gap filling and smoothing processes did not have any significant impact on forecast model as shown in (Barrett et al., 2020). Since our target variable was computed from the long-term minimum and maximum NDVI, the additional variables were also converted to anomalies by subtracting their long-term means to produce soil moisture anomaly and precipitation anomaly. The persistence within individual variables was enhanced by computing with three months (12 weeks) rolling averages to derive three-month VCI (VCI3M), three-month precipitation (P3M) and three-month soil moisture (SM3M). Finally, the precipitation and soil moisture data were standardised to eliminate any associated units of measurements and avoid the dominance of certain variables. This was done by subtracting their mean and dividing it by the standard deviation.

---

[2]Visit this link (http://2016africalandcover20m.esrin.esa.int/) to learn more

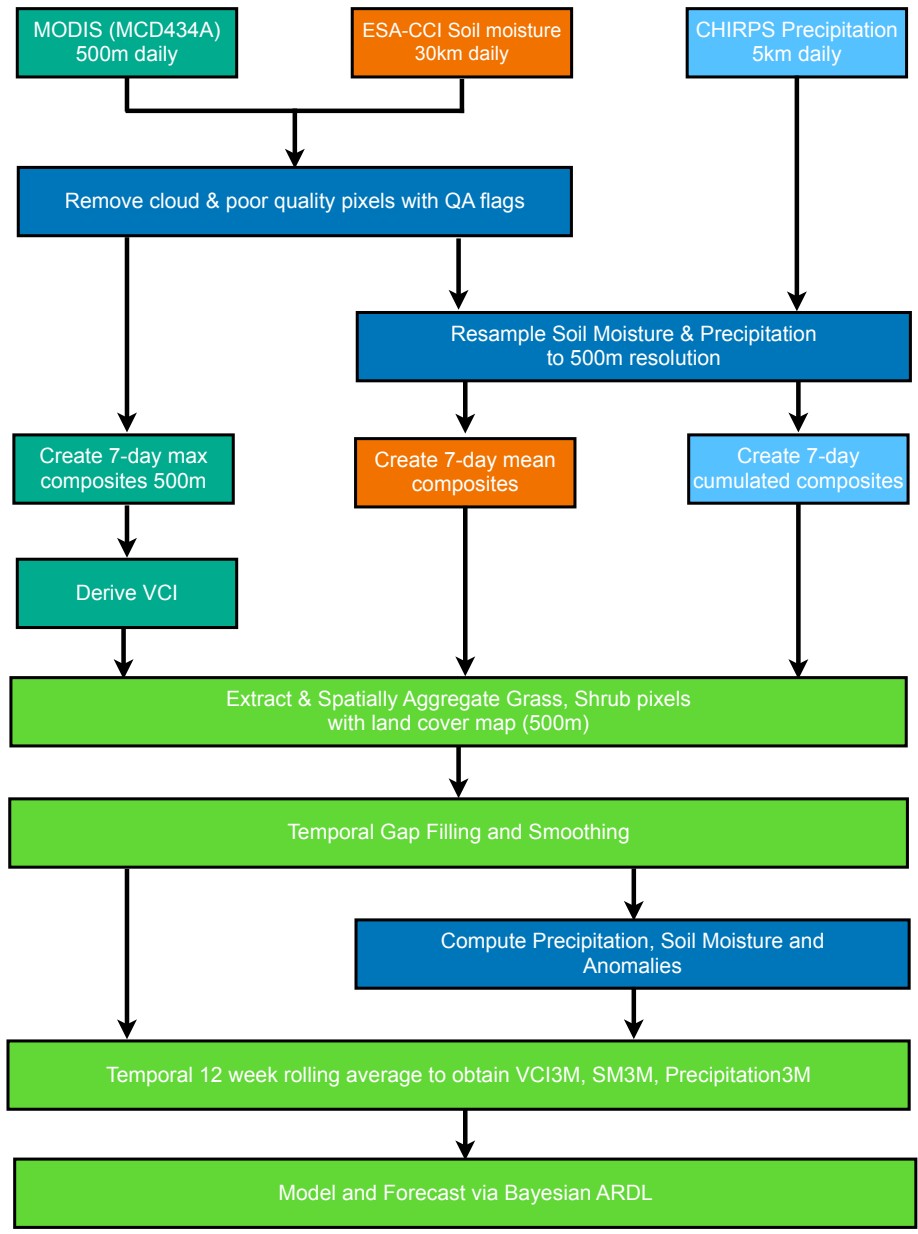

**Figure 2.** A flow chart showing data prepossessing and modelling

## 3.2 Drought Model and Forecasting

The AR method used in (Barrett et al., 2020) was used as a baseline model for this study. The AR(q) model, with $q$ being the number of lags, used historical values of VCI3M in the linear regression model to forecast future VCI3M. The AR(q) model was defined as:

$$D_{t+n} = \alpha_0 + \sum_{i=0}^{q} \beta_d D_{t-q} + \epsilon_{t-q} \tag{3}$$

where the $D_{t+n}$ is the VCI3M at $n$ lead time ahead and $D_{t-q}$ are the lags (0, to $q=3$) of past VCI3M values. $\alpha_0$ represents the intercept and $\epsilon_{t-p}$ the error term.

The results from the AR baseline model was compared to the output of the Auto-Regressive Distributed Lag (ARDL)proposed in this paper. The Auto-Regressive Distributed Lag (ARDL) modelling approach used for this work is a generalised form of AR method mainly used for multivariate time series analysis. The method enables the variable of interest (dependent variable) to be modelled as a function of its lags and that of additional explanatory variables (independent variable) (Gujarati, 2003). An ARDL*(p,q)*, consists of *p*, which represents the number of lags of the independent variable and *q*, which is the auto-regressive part of the model, represents the number of lags of the dependent variable. This approach has been extensively used in the field of economics and modelling the effect of climate and environmental variables on vegetation (Lei Ji and Peters, 2004; Ji and Peters, 2005).

For this study, however, parameter estimation for the ARDL was implemented within a Bayesian framework instead of using maximum likelihood methods based on Ordinary Least Squares (OLS). The Bayesian framework enables the incorporation of domain knowledge about the parameters through the use of informative priors. The model parameters, with this approach, are inferred using the Markov Chain Monte Carlo (MCMC) (Neal, 1993) sampling algorithm. The sampling process generates posterior probability distribution of the model parameters. As a consequence, we get a full probability distribution of forecast values for all lead time, which makes it easy to quantify forecast uncertainty for making informed decisions (Martin, 2018; Lambert, 2018).

The MCMC is a well-established sampling algorithm used for parameter inference in Bayesian models. However, Asaad and Magadia (2019), outlined some of its limitations and recommended the use of the Hamiltonian Monte Carlo (HMC) (Hoffman and Gelman, 2014), an improved variant of the traditional MCMC algorithm which is based on Hamiltonian dynamics and converges faster to a global minimum for models with high dimensional parameter space. (Robert et al., 2018). Parameter inference for this work was done with the No-U-Turn Sampler (NUTS)(Hoffman and Gelman, 2014) version of HMC implemented with PyMC3 (Salvatier et al., 2016) Python package.

The Bayesian ARDL model used for forecasting VCI3M with lagged P3M, and S3M is defined as:

$$D_{t+n} = \alpha_0 + \sum_{i=0}^{q} \beta_d D_{t-q} + \sum_{i=0}^{p} \theta_p P_{t-p} + \sum_{i=0}^{p} \delta_s S_{t-p} + \epsilon_{t-p} \tag{4}$$

where $D_{t+n}$ is the drought index at $n$ lead time, $D_{t-q}$ are the lags (0, to $q$) of drought indicator (Dependent variable). $P_{t-p}$, $S_{t-p}$ represent the lags 0, to $p$, for precipitation, and soil moisture respectively. $\alpha_0$ is a constant representing the intercept and $\beta_d$, $\theta_p$, and $\delta_s$ are the regression coefficients of the input variables with $\epsilon_{t-p}$ being the error term which is assumed to be Gaussian.

Equation (4) can be re-written as:

$$D_{t+n} = \alpha + \sum_{i=0}^{i} \beta_i X_{t-i} + \epsilon_{t-i} \tag{5}$$

where $n$ is the lead time, $\beta_i$ are the model parameters and $X_{t-i}$ represent the lagged input variables in equation 4.

The Bayesian approach makes explicit the prior beliefs about model parameters, which are then updated given some new data via the likelihood function, to give the posterior probability distribution.

Parameter inference with the Bayesian framework is based on Bayes' theorem via the equation below:

$$P(\theta|X_t) = \frac{P(X_t|\theta).P(\theta)}{P(X_t)} \tag{6}$$

where $\theta$ is the model parameter, $X_t$ represents $D_{t-q}, P_{t-p}, S_{t-p}$, $P(\theta|X_t)$ is the posterior or the probability of our model parameters given our data $X_t$, $P(X_t|\theta)$ is the likelihood or the probability of the data given the parameters, $P(\theta)$ is our prior belief about the parameters. $P(X_t)$, known as the evidence, is a normalisation term that represents the probability of the data. The term is intractable and usually ignored (Lambert, 2018; McElreath, 2016). Thus the equation (6) for Bayes' theorem is re-written as:

$$P(\theta|X_t) \propto P(X_t|\theta).P(\theta) \tag{7}$$

To put the ARDL model (equation 5) in the context of equation 7, the likelihood function $P(X_t|\theta)$ is written as:

$$P(X_t|\alpha, \beta_i, \sigma) \sim N(\alpha + \sum_{i=0}^{i} \beta_i X_{t-i}, \sigma_{t-i}) \tag{8}$$

Computing equation 7 requires very complex integrals (Lambert, 2018) thus the HMC sampling algorithm (Hoffman and Gelman, 2014) was used for estimating the model parameters.

The prior $P(\theta)$ for the model's regression coefficients are assumed to be Gaussian $P(\theta) = N(\mu, \sigma)$. with $\mu$ set to 0 to allow inferred parameters to have both positive and negative values and a weakly informative $\sigma$ of $0.5$ as a regularization prior. This was done to avoid the approximation of unreasonable parameters. (Martin, 2018). The weakly informative $\sigma$ of $0.5$ was chosen after a grid search to select optimal parameters.

### 3.3 Selecting optimal lags and forecasting

A full grid search was done with various combinations of $p$ and $q$ values for dependent and independent values to select the optimal $p$ and $q$ for the BARDL model. The Akaike Information Criterion (AIC) (Akaike, 1998) equation (9) and the $R^2$

(Equation 10) metric were used as the score criteria to choose the optimal lags. AIC enables model selection by determining the model that best fits the data. The model with lowest AIC value is preferred. Whereas the $R^2$ score explains how much variation in the observed data could be explained by the model. Valid $R^2$ scores range between 0&1 where models with scores close to 1 are considered more accurate. The search was done on lag values ranging from 1 to 16 weeks. The best AIC and
$R^2$ scores varied for different lag combinations and also for each county. However, across all counties an optimal AIC and $R^2$ scores were obtained when all input variables were set to a lag of 6 weeks. The AIC scores are derived as follows:

$$\text{AIC} = 2K + n\log(\frac{RSS}{n}) \tag{9}$$

where the $RSS$ is the residual sum of squares error, $n$ is the number of data points and $K$ is the number of estimated parameters.

The $R^2$ scores are derived as follows:

$$R^2\text{-score} = 1 - \frac{\sum_i(y_i - f_i)^2}{\sum_i(y_i - \bar{y})^2}, \tag{10}$$

where the $y_i$ are the observed data, $\bar{y}$ is the mean of the observed data and the $f_i$ are the forecasts.

Forecasting with the BARDL was done using the direct multi-step forecast approach, where separate models are fitted for $n$ step ahead forecasts (Ben Taieb et al., 2010; Ben Taieb and Hyndman, 2014). To fit the model for $n$ steps ahead, the data was restructured to offset values of the dependent ($D_{t+n}$), $n$ weeks from lag 0 $X_{t-0}$ for all input variables. A rolling window cross-
validation approach (Hyndman and Athanasopoulos, 2018) was used for model training and forecasting. With this approach, the data is divided into chunks of 500 data points, for each chunk, 400 data points are used to train the model and remaining 100 data points held-out for prediction. The observed values from held-out data and mean forecast distribution from the Bayesian model were then used to evaluate the model skill.

### 3.4   Forecast skill assessment

The performance of the models was assessed by measuring the *accuracy*, i.e. how well the forecasts agree with the observations and the *precision*, i.e. the quoted uncertainty and the accuracy of that uncertainty.

The model *accuracy* was evaluated with the $R^2$ (equation 10) and Root Mean Squared Error (RMSE) (equation 11). The RMSE measures the mean deviation between the observed and forecast values.

$$\text{R}MSE = \sqrt{\frac{\sum_{i=1}^{n}(y_i - f_i)^2}{n}} \tag{11}$$

where the $y_i$ are the observed data, $f_i$ are the forecasts and $n$ the total number of data points.

The *precision*, was quantified with the Prediction Interval Coverage Probability (PICP) and the Mean Prediction Interval Width (MPIW) (Pang et al., 2018) were also computed. The MPIW measures the average width between the upper ($u(D_i)$) and lower bound $l(D_i)$ of a proportion of forecast distribution ($n$ weeks ahead ) defined by a chosen prediction interval (e.g. 95%). See figure 3 for an illustration.

$$MPIW_{t+n} = \frac{1}{N}\sum_{i=1}^{N}|u(D_i) - l(D_i)|. \tag{12}$$

The PICP shows the percentage of time the observed variable lies within the credible interval of the forecast distribution and is derived as follows:

$$PICP_{t+n} = \frac{1}{N}\sum_{i=1}^{N}c_i \tag{13}$$

where $N$ represent the number of predicted samples and $c_i$ is either 0, or 1. If the observed drought target variable falls
within the upper and lower bound of the forecast distribution ($n$ weeks ahead) then $c_i = 1$; else $c_i = 0$ (figure 3) if otherwise.

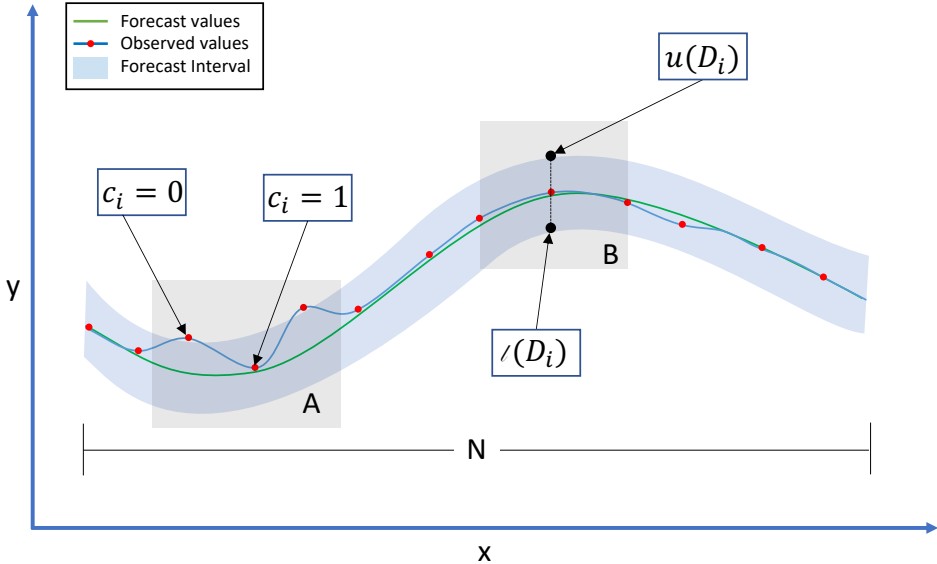

**Figure 3.** A diagram illustrating how the Prediction Interval Coverage Probability (PICP) (A) and the Mean Prediction Interval Width (MPIW) (B) are computed.

The goal is to minimize the MPIW while maintaining a high PICP value. A high PICP value (0.90 to 0.99) indicates that the observed values lie within the forecast distribution and a low MPIW value indicates a more precise forecast (Su et al., 2018). For the AR model, the confidence interval used to derive its PICP and MPIW was computed with the forecast RMSE and z-score of 1.96 representing the 95% confidence level of a standard normal distribution. This was done to permit its comparison
to the output of the full BARDL model.

The contribution of the individual lagged input in the ARDL model were also measured by computing their percentage relative importance via the Relative Weight Analysis method (Tonidandel and LeBreton, 2011). With this approach, the input

variables are initially transformed into orthogonal variables. Through an iterative process, each orthogonal variable is added to a linear regression model and the change in $R^2$ score for each iteration is measured and expressed as a percentage of the total $R^2$ score.

The Receiver operating characteristic (ROC) curve was also plotted to see how well the model forecasts a drought event given a threshold. The ROC shows the probability of a forecasted event being true (True Positive Rate (TPR)) against the chance of that predicted event being false alarm (False Positive Rate (FPR)) at different thresholds. The Area Under the Curve (AUC), quantifies the ability of the forecast model to distinguish between drought events (Wilks, 2006; Bradley, 1997).

The forecast distribution from our BARDL model enabled the computation of forecast probabilities given some drought thresholds. The forecast probability of a drought event was computed from the full forecast distribution from our posterior at a drought threshold of VCI<35. The model's skill at accurately forecast these probabilities was assessed by plotting and analysing a Reliability Diagram and Sharpness. The reliability diagrams were plotted by using same threshold of VCI<35 to initially convert observed VCI3M data at a lead time into binary events where 0 indicates a 'No Drought' and 1 indicate a 'Drought' event. The forecast probabilities and observed binaries were binned into standard intervals and plotted as a joint distribution of forecast probabilities and the relative frequency of the true observed drought event (where observed binaries = 1). The sharpness plots, on the other hand, are histograms of drought occurrences in each probability bin.

The Reliability Diagram shows how well forecast probabilities for a given drought event agreed with its corresponding observed event while the Sharpness shows the frequency of a forecasted drought event. (WWRP, 2009; Wilks, 2006).

## 4  Results

### 4.1  Forecast accuracy

AR modelling approach had proved to be skilful for short-range (2 to 6 weeks lead time) VCI3M forecasts (Barrett et al., 2020). However, the goal of this study was to extend the forecast range beyond 6 weeks while maintaining high accuracy by using the BARDL model and considering the effect of exogenous factors like precipitation and soil moisture. The results shown in this section are for 6 to 12 weeks lead time for the BARDL models and with the AR modelling as a comparative baseline. All the evaluation metrics for the BARDL outputs were computed with the mean forecast distributions from our Bayesian models.

The contour plots in Figure 4, shows a joint distribution (Scatter Plot) of the observed VCI3M and forecasted VCI3M at 6, 8, 10 & 12 weeks for both AR and the BARDL models. The coloured contour lines represent the bins of the joint histograms and for each plot, the correlation ($r$), RMSE and $R^2$ were computed. Overall, the results from the BARDL model showed a roughly 2-week gain in the performance metrics. For instance, $R^2$ score for the AR model at 6 weeks is equivalent to $R^2$ score at 8 weeks lead time for the BARDL models. This pattern can be seen across all forecast ranges for the RMSE as well.

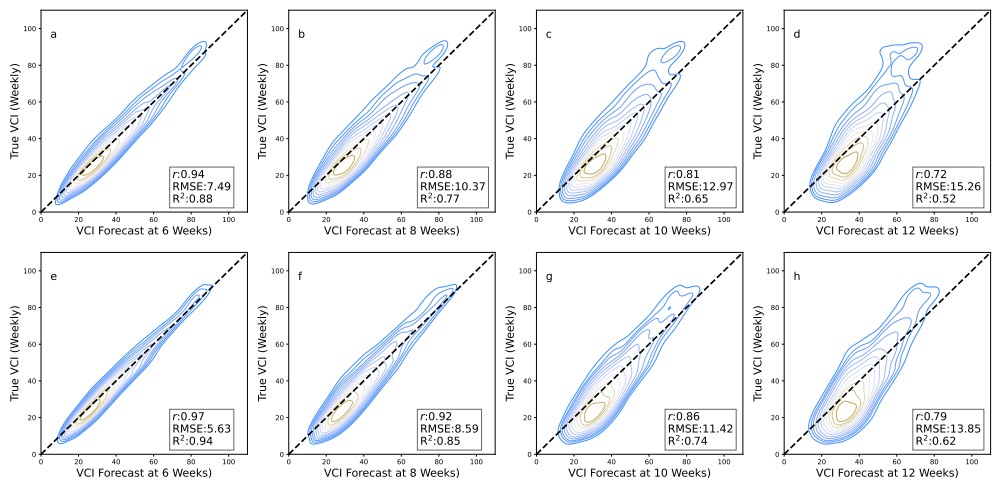

**Figure 4.** Contour plots showing VCI3M forecast against True VCI3M. Plots (a,b,c,d) shows the results from the AR method with VCI3M only, (e,f,g,h) shows the overall results for BARDL modelled with lags of VCI3M plus lags of Precipitation (P3M) and Soil Moisture (S3M) Anomalies for 6, 8, 10 and 12 weeks lead time for all counties. The contour lines (ellipses) seen in each plot indicate the various bins that make up the joint histogram plots between the forecasted and observed VCI3M values. Contour lines with a yellow shade indicate values with stronger correlation between the observed and predicted values.

The performance metrics for the BARDL model in comparison to the AR model are shown in figure 5. This shows a significant improvement in performance at the same lead-time and, as a consequence, similar performance in the BARDL models is seen 2 weeks ahead of the AR models.

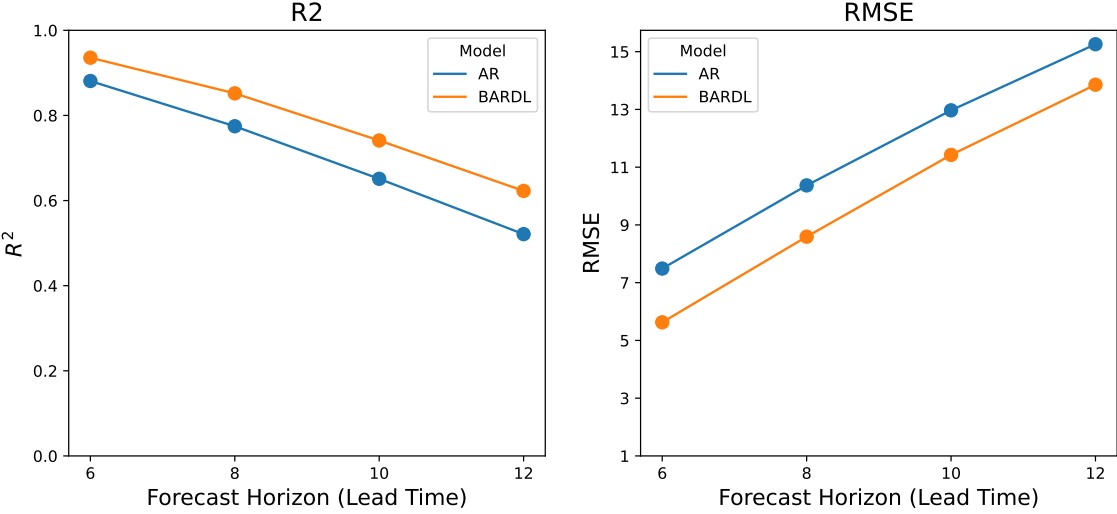

**Figure 5.** Performance metrics used to measure model accuracy as a function of forecast lead time. $R^2$(Left), RMSE (Right).

Table 2 shows the $R^2$ scores for 6 to 12 weeks forecasts for AR and BARDL models at the county level for arid and semi-arid regions. Just as observed in the contour plots, significant improvements are seen from 6 to 10 weeks lead time across all counties. In an arid county like Mandera, the $R^2$ improved from 0.84, 0.72 and 0.58 using AR to 0.93, 0.84 and 0.73 using BARDL for 6, 8 and 10 weeks lead times respectively. Kitui in the semi-arid region also showed an improvement in $R^2$ score from 0.84, 0.71 and 0.57 to 0.91, 0.81 and 0.67 for weeks 6, 8 and 10 respectively. Overall the BARDL method demonstrated better results compared to the AR across all counties.

Further evaluation of forecasts based on Kenya's long rain (March, April, May (MAM)) and short rain (October, November, December (OND)) seasons (Camberlin and Wairoto, 1997) also showed even better $R^2$ score for longer range forecast in MAM season compared to the OND for the BARDL model as seen in figure 6 as well as in figure D1 in appendix C.

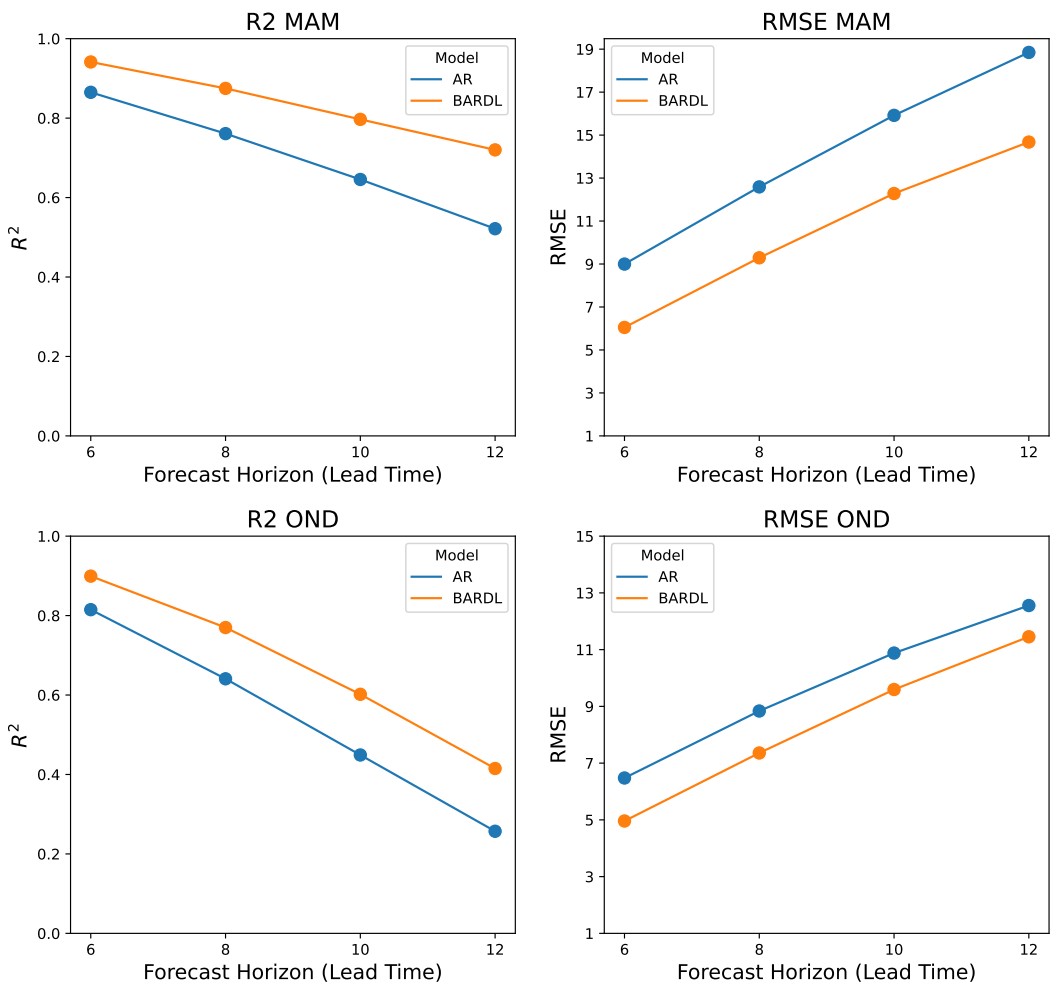

**Figure 6.** Performance metrics used to measure model accuracy as a function of forecast lead time for MAM and OND Season.

The $R^2$ scores for the AR model in the MAM season however dropped significantly compared to the OND season.

**Table 2.** $R^2$ scores (6 to 12 weeks lead times) for AR modelled with lags of VCI3M only, BARDL modelled with lags of VCI3M with Precipitation (P3M) and Soil Moisture (SM3M) for arid and semi-arid counties. The mean $R^2$ values in this table do not correspond to $R^2$ values in the figure 4 because figure 4 shows the overall scores for all counties, while the table shows the scores for the separate Arid and Semi-Arid zones.

| | County | AR | | | | BARDL | | | |
|---|---|---|---|---|---|---|---|---|---|
| | | 6 | 8 | 10 | 12 | 6 | 8 | 10 | 12 |
| Arid Counties | Garissa | 0.88 | 0.78 | 0.66 | 0.54 | 0.92 | 0.83 | 0.71 | 0.60 |
| | Isiolo | 0.89 | 0.79 | 0.67 | 0.55 | 0.95 | 0.88 | 0.77 | 0.66 |
| | Mandera | 0.86 | 0.74 | 0.60 | 0.46 | 0.93 | 0.84 | 0.73 | 0.63 |
| | Marsabit | 0.91 | 0.81 | 0.69 | 0.54 | 0.96 | 0.90 | 0.80 | 0.68 |
| | Samburu | 0.88 | 0.75 | 0.59 | 0.43 | 0.95 | 0.87 | 0.75 | 0.62 |
| | Tana-River | 0.85 | 0.75 | 0.64 | 0.53 | 0.92 | 0.83 | 0.72 | 0.62 |
| | Turkana | 0.90 | 0.79 | 0.65 | 0.50 | 0.96 | 0.89 | 0.79 | 0.65 |
| | Wajir | 0.82 | 0.69 | 0.55 | 0.42 | 0.91 | 0.82 | 0.71 | 0.61 |
| | **Mean** | **0.87** | **0.76** | **0.63** | **0.50** | **0.94** | **0.86** | **0.75** | **0.63** |
| | *Std. Dev.* | *0.03* | *0.04* | *0.04* | *0.05* | *0.02* | *0.03* | *0.03* | *0.03* |

| | | 6 | 8 | 10 | 12 | 6 | 8 | 10 | 12 |
|---|---|---|---|---|---|---|---|---|---|
| Semi-Arid Counties | Baringo | 0.92 | 0.83 | 0.70 | 0.56 | 0.95 | 0.86 | 0.74 | 0.60 |
| | Kajiado | 0.90 | 0.80 | 0.69 | 0.57 | 0.96 | 0.90 | 0.81 | 0.71 |
| | Kilifi | 0.84 | 0.72 | 0.60 | 0.48 | 0.88 | 0.76 | 0.62 | 0.49 |
| | Kitui | 0.84 | 0.70 | 0.56 | 0.43 | 0.92 | 0.81 | 0.68 | 0.53 |
| | Laikipia | 0.93 | 0.85 | 0.73 | 0.59 | 0.97 | 0.91 | 0.81 | 0.67 |
| | Makueni | 0.84 | 0.72 | 0.59 | 0.46 | 0.93 | 0.83 | 0.71 | 0.59 |
| | Meru | 0.83 | 0.67 | 0.49 | 0.33 | 0.92 | 0.81 | 0.67 | 0.52 |
| | Narok | 0.85 | 0.74 | 0.60 | 0.45 | 0.92 | 0.81 | 0.67 | 0.50 |
| | Nyeri | 0.90 | 0.81 | 0.68 | 0.54 | 0.93 | 0.85 | 0.73 | 0.60 |
| | Taita-Taveta | 0.86 | 0.74 | 0.60 | 0.47 | 0.92 | 0.81 | 0.69 | 0.59 |
| | Tharaka-Nithi | 0.81 | 0.64 | 0.45 | 0.28 | 0.83 | 0.63 | 0.39 | 0.17 |
| | West-Pokot | 0.91 | 0.82 | 0.69 | 0.54 | 0.95 | 0.86 | 0.72 | 0.57 |
| | **Mean** | **0.87** | **0.75** | **0.62** | **0.48** | **0.92** | **0.82** | **0.69** | **0.54** |
| | *Std. Dev.* | *0.04* | *0.06* | *0.08* | *0.09* | *0.04* | *0.07* | *0.10* | *0.13* |

## 4.2 Uncertainty Analysis (PICP and MPIW)

The PICP and MPIW for a $95\%$ forecast confidence interval were computed for each lead time for both the AR and BARDL models. In figure 7, the time series plots show that the observed VCI3M values lie within the 95% forecast interval between 90–94% of the time across all lead times for both the BARDL and AR models. However, lower values of MPIW demonstrate that the BARDL provided more precise forecasts. Appendix A, tabulates PICP and MPIW for 6 to 12 weeks forecasts for the AR and BARDL models for all counties (Table A1).

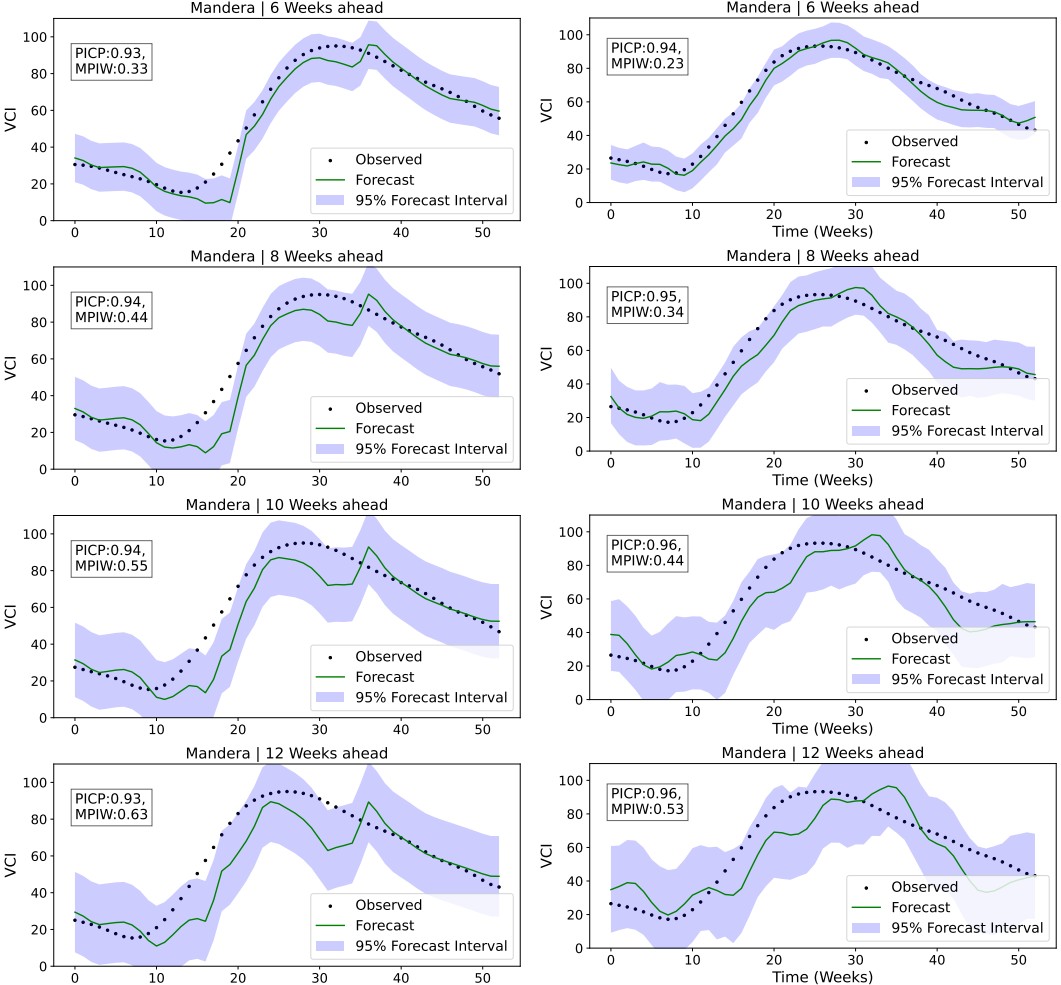

**Figure 7.** Time series plot showing uncertainty for 6, 8, 10, 12 weeks lead time for Mandera county. Plots on the left side are from the AR model and plots to the right are BARDL. The PICP and MPIW for the other counties can found in Appendix A. The forecast line (green line) represents the mean values of the BARDL outputs.

## 4.3 Drought Events ROC Curve

The Receiver Operating Characteristic (ROC) curve (figure 8) illustrates how well the model can discriminate drought events. Drought events are forecasted when the predicted VCI3M drops below a threshold and are deemed correct if the observed VCI3M is below 35 (moderate to severe drought) (Klisch and Atzberger, 2016). The ROC curve and AUC metric for the BARDL model also demonstrated an improvement over the AR model. The points plotted on the curve represent the TPR and FPR where VCI3M<35. This indicates that when the AR model (Dotted curve), forecasts a drought condition (i.e VCI3M<35) for 6 weeks ahead, the probability of it being true is 86% with a FPR of 9%. Whereas a forecast by the BARDL model (Solid curve) at the same 6 weeks had a TRP of 89% and a FPR of 7%. The improvements with the BARDL model were mainly seen in the TPRs (6 to 10 week lead time) for the BARDL model while the FPR remained almost the same. The improvements seen in the ROC curves in figure 8 are however not reflective of the distinct improvement seen in figure 5. The observed difference was because whereas the $R^2$ and RMSE are comparing the explained variances and deviation between the observed and forecast VCI3M, the ROC is mainly assessing the skill of both models at predicting drought occurrence at the VCI3M<35 threshold.

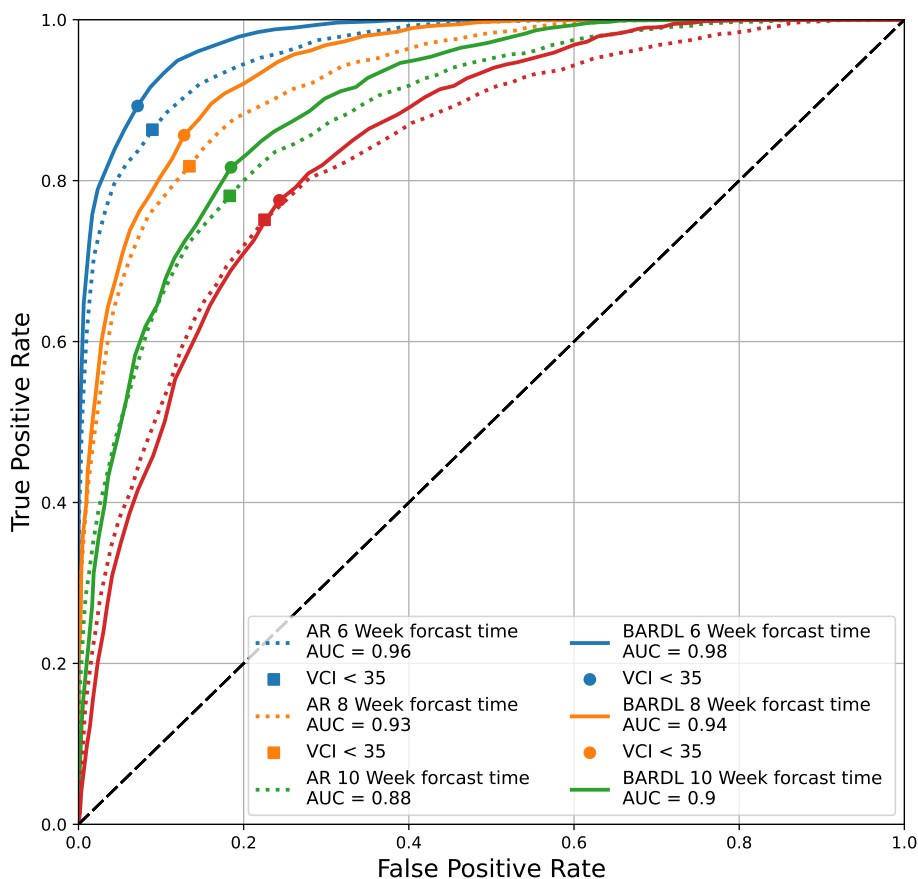

**Figure 8.** ROC Curve showing True Positive Rate (TPR), False Positive Rates(FPR) and AUC for 6,8,10,12 weeks for both AR (Dotted line) and BARDL (Solid line) forecasts. The VCI3M < 35 threshold is plotted as points on the lines. The AUC scores for models was above 80% indicating both models were effective at identifying drought events.

### 4.4 Forecast Reliability

Using the Bayesian approach also enabled the computation of forecast probabilities for a given drought event (No Drought Condition – VCI3m>35 or Drought Condition – VCI3M<35). To assess the skill for forecasting drought probabilities, we used the reliability diagrams in figure 9. The plot shows a joint distribution between the forecast probabilities in bins and the frequencies of the observed drought events that fall in those bins. For each lead time, the sharpness histogram which shows the frequency at which an event is forecasted (WWRP, 2009) are also plotted. The reliability of a perfect model would follow the line $y = x$ which has been represented by a dashed line in figure 9. The closer a model is to this dashed line, the more reliable it is. Figure 9 shows the reliability for drought events (VCI3M<35) in arid counties, from the forecast skill assessment, our BARDL model indicates that when we forecast a 'Drought event' with a probability between 80% to 100% for 6 week lead time, it corresponds with the observed drought events about 88% to 99% of the time. This diagram, however, showed that the

forecast probabilities from the BARDL model do not always agree with observed event indicating situation known as *'under forecasting'* (Wilks, 2006).

In terms of the model's sharpness, it can be seen that most of the drought events forecasted by the BARDL model have a probability between 90% to 100%. The peak at the 0% to 10% bin of the sharpness plot shows the frequency of 'No Drought' forecasts in the arid counties. This indicates the likelihood of the model missing some drought events especially from 8 weeks lead time and beyond.

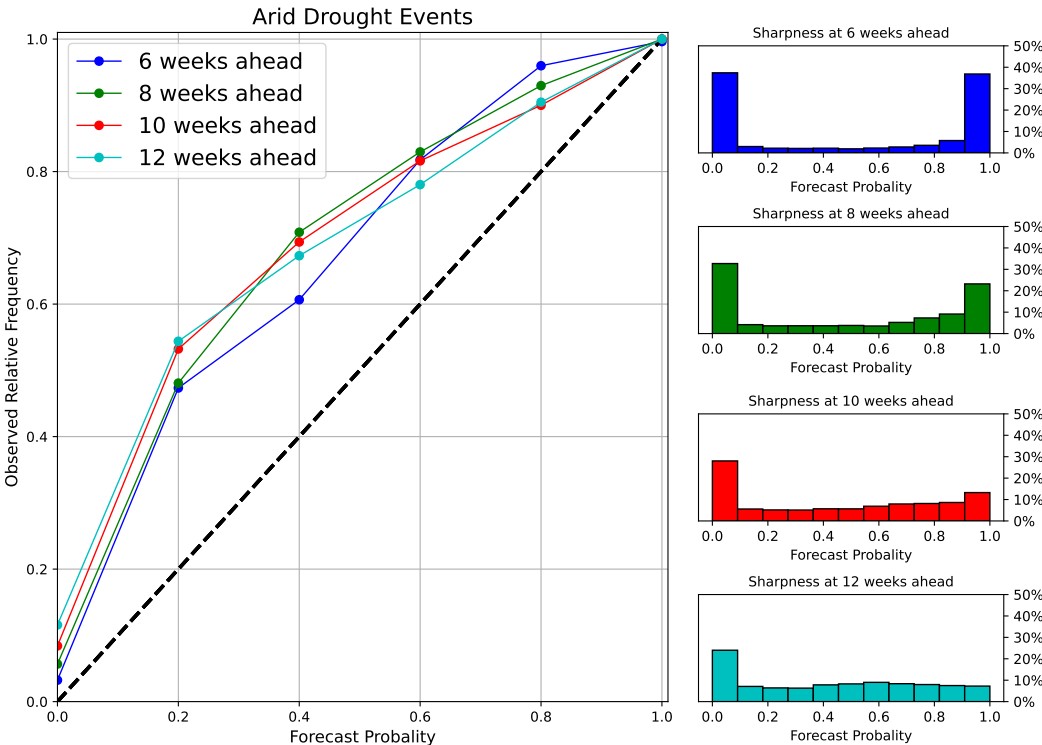

**Figure 9.** Reliability diagram showing forecast probability and their corresponding observed frequencies for 6, 8, 10, 12 weeks lead time together with their corresponding sharpness plots for drought events (VCI3M< 35) in the arid and semi-arid counties

Another key observation of the reliability diagrams for the MAM and OND seasons (figure 10) showed the model was sharp
at identifying more drought events in the short-rain season compared to the long-rain season.

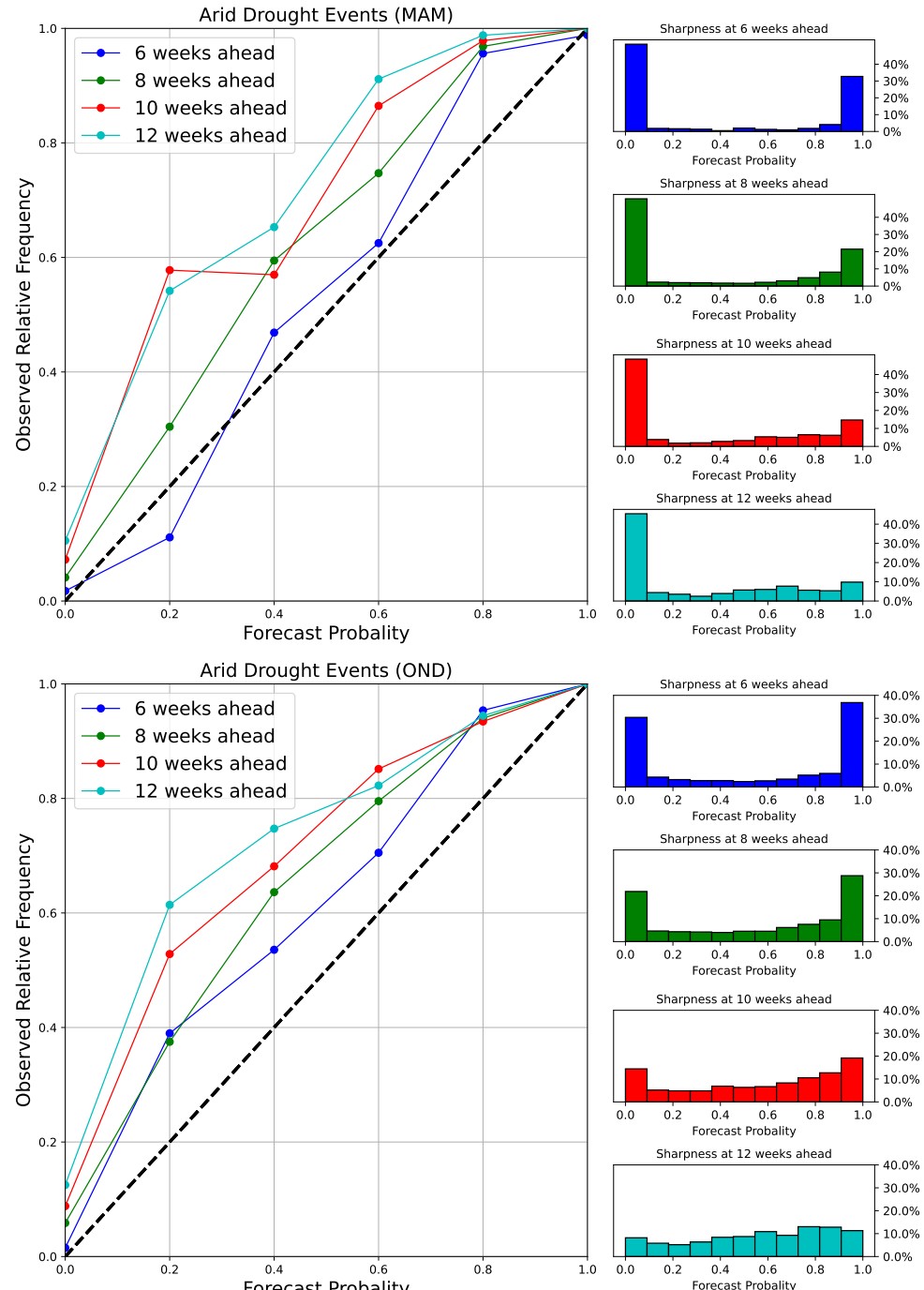

**Figure 10.** Reliability diagram showing forecast probability and their corresponding observed frequencies for 6, 8, 10, 12 weeks lead time together with their corresponding sharpness plots for drought events (VCI3M< 35) MAM and OND

## 4.5 Relative Importance

Figure 11 shows the cumulative percentage relative importance for the lags of VCI3M, P3M anomaly and SM3M anomaly. The lags of VCI3M contributes the most for shorter lead time and decreases longer lead times. The precipitation anomaly also contributes significantly to future VCI3M and its relative importance increases with increasing forecast lead times. The relative importance of soil moisture, although it varies less across various lead times, also contributes significantly. Detailed plots of the relative importance for individual lag contribution for each arid and semi-arid county in figure B1 (Appendix B). Figure B1 also showed that input variables contributed mostly at lag 0. A critical look at these plots also showed that VCI3M responds better to precipitation anomaly in most arid counties like Turkana and Wajir compared to semi-arid counties like Kitui and West-Pokot. The relative importance of the lagged exogenous factors for different MAM and OND seasons also confirms the reliance of future VCI3M on precipitation anomalies as seen figure C1 in appendix D.

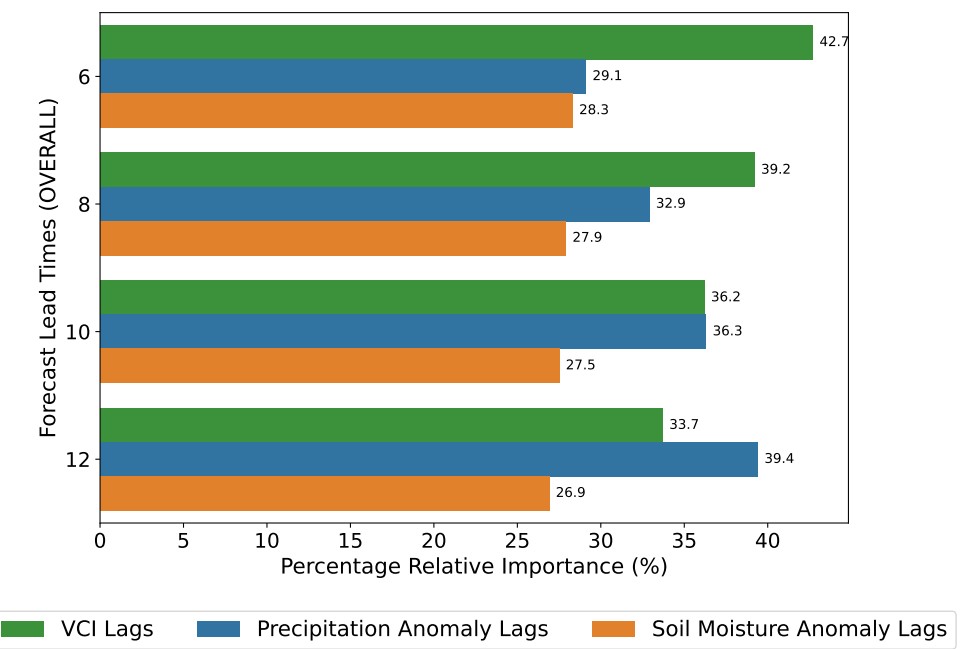

**Figure 11.** Bar plots showing the cumulative (All lags) relative importance of additional variables to the VCI3M forecast for all counties

## 5 Discussion

Our BARDL model, which incorporated lagged precipitation and soil moisture anomalies as exogenous factors, exhibited an approximately 2-week gain in forecast range compared to the baseline model. These gains could be been in the figure 5 and table 2 where, for instance, an $R^2$ score obtained at a 6 weeks forecast for AR model was equivalent to an $R^2$ score at 8 weeks for the BARDL model. Forecasts from the BARDL were mostly driven by the variables at lag 0 (See figure B1). However, the

collective contribution of the additional lags substantially improved the forecast ranges . Finally, the skill assessment based on forecast probabilities indicated a good separation between No-Drought and Drought events.

The results from the model evaluation revealed a strong persistence within soil moisture and VCI3M, a property that enables future values to be inferred from their past values (AghaKouchak, 2014). These could be seen from the lag contribution of

325 soil moisture and precipitation in figures 11, B1 and C1. Despite this inherent persistence in the VCI3M, it still required the information from additional biophysical factors to improve its forecast range as seen in figure B1 and the overall performance of the BARDL model. Another interesting observation from figure 11 also showed that VCI3M responded very slowly to short term moisture anomalies (Quiring and Ganesh, 2010; Vicente-Serrano, 2007). From a spatial perspective, both models (AR & BARDL) gave a higher forecast $R^2$ score in the arid areas compared to the semi-arid areas. This was more significant for the

330 BARDL model.

The performance ($R^2$ scores) of the BARDL model during the long rain (MAM) seasons indicated that although VCI3M responds slowly to short term moisture levels, the impact of precipitation and soil moisture on vegetation condition is very vital. However the low $R^2$ scores seen for the AR model in the MAM season, could be attributed to the absence of information from the moisture levels (precipitation and soil moisture) in the AR model. The reliability plots for MAM and OND seasons

also showed that the contribution of the lagged soil moisture anomalies during the OND seasons also increased compared to the MAM season. This was an indication that during the short rain season, vegetation condition is controlled mostly by soil moisture. The sharpness plots in figure 9 also indicated that the model was generally sensitive to identifying drought events more frequently till eight weeks ahead. However, when it comes to forecasting drought events, a much higher frequency is seen during the OND season (see figure 10. This is expected since there are fewer rains in the OND seasons.

The uncertainty analysis from the PICP showed the observed VCI3M values were within the upper and lower bounds of the forecast distribution about 90% of the time indicating a low forecast uncertainly. The MPIW revealed that the forecast intervals were generally slightly narrower for the BARDL model compared to the AR model. Overall, higher PICP values were seen for AR however, the the PICP and MPIW values for BARDL model are assumed to be a true representation of the forecast error since they were computed from forecast distribution.

Aside from the significant improvements in the forecast range and precision, the strength of our model hinges on the fact that we implemented it in a Bayesian context. Using the Bayesian approach generates a full posterior probability distribution of forecasted VCI3M values which gave us the power to easily gain insight into the uncertainty of forecasted VCI3M values (Lambert, 2018). It also allowed the computation of probabilistic forecast of specific drought events (e.g. VCI3M falling in a particular range) (Wilks, 2006). For our target end-users and stakeholders like the NDMA, using the Bayesian model

proposed in this paper as part of their EWS will enable them to confidently report on drought events. Also, policymakers and administrators of disaster relief organisations based on the forecast-based finance initiatives (Coughlan de Perez et al., 2015), can make better decisions and prioritise which drought alarms to act on. This will help with the efficient management of funds. The soil moisture data though retrieved via a combination of remote sensing and a soil moisture model (Gruber et al., 2019) also proved useful for drought monitoring and forecast. The extensive model skill assessments done here shows that our Bayesian

ARDL approach not only performs better compared to results from previous studies (Barrett et al., 2020) but also, the BARDL model, by design, provides additional uncertainty information for better decision making.

Although we have shown that we can extend forecast ranges with the added variables, the limitations to work include, the availability of soil moisture data. The ESA CCI Soil Moisture products used in this paper are released annually and are also a year behind. Thus they cannot currently be used for producing real-time forecasts. Another limitation was the use of the 2016 ESA Sentinel 2 land cover map for sampling grassland and shrub pixels across an 18-year period. Even though the land cover product accurately depicted areas with grassland and shrubs, pixel values from regions with significant land cover changes over time may be misleading.

## 6 Conclusion and Future Work

In this paper we increased the range of VCI3M forecasts, using additional lagged information from P3M and S3M anomalies. The VCI3M used here was derived from the 12-week rolling mean of VCI, as used by Kenya's NDMA for monitoring and reporting agricultural droughts occurrences. Key highlights in the paper include; the improvement in the forecast range of VCI3M using lagged information from precipitation (P3M) and soil moisture (S3M) by approximately 2 weeks compared to previous work that was based AR model. Secondly, modelling within the Bayesian framework also gave the added advantage of easily assessing model uncertainty and probability of a drought events. Results showed the our proposed model forecasted VCI3M at higher accuracy at a longer range during the MAM season and was also more sensitive to drought events during the OND season.

The forecast-based finance initiatives aimed at monitoring agricultural drought indicators and their impact on livelihoods should consider Bayesian approaches to enable better decision making. We would also recommend that soil moisture data be made available sooner and promptly to enable near real-time forecasting of vegetation condition via our proposed method.

The disparity in model performance between arid and semi-arid regions points to the fact that the differences in climate and vegetation land use and land cover (LULC) should also be considered when developing such forecast models. A natural expansion of our BARDL model would be to simultaneously explore and model for spatial variations like LULC in a county or any region of interest via a hierarchical modelling approach. Doing this will give us the advantage of pooling information between spatial variations, whilst still allowing flexibility between them.

*Code and data availability.*  Link to Data and Code repository https://github.com/edd3x/Bayesian-ARDL.git

**Appendix A:  A table showing the PICP and MPIW (in brackets) estimates for the arid and semi-arid counties**

**Table A1.** The PICP and MPIW (in parenthesis ) estimates for the all arid and semi-arid counties.

| | County | AR Model | | | | BARDL Model | | | |
|---|---|---|---|---|---|---|---|---|---|
| | | **6** | **8** | **10** | **12** | **6** | **8** | **10** | **12** |
| Arid Counties | Garissa | 0.93 (0.33) | 0.94 (0.46) | 0.94 (0.57) | 0.93 (0.67) | 0.86 (0.21) | 0.84 (0.31) | 0.83 (0.39) | 0.81 (0.46) |
| | Isiolo | 0.93 (0.29) | 0.93 (0.4) | 0.93 (0.51) | 0.93 (0.6) | 0.94 (0.18) | 0.92 (0.27) | 0.9 (0.36) | 0.89 (0.43) |
| | Mandera | 0.93 (0.33) | 0.94 (0.44) | 0.94 (0.55) | 0.93 (0.63) | 0.94 (0.23) | 0.95 (0.34) | 0.96 (0.44) | 0.96 (0.53) |
| | Marsabit | 0.92 (0.25) | 0.91 (0.36) | 0.93 (0.46) | 0.94 (0.56) | 0.93 (0.15) | 0.9 (0.23) | 0.88 (0.32) | 0.88 (0.38) |
| | Samburu | 0.95 (0.26) | 0.94 (0.37) | 0.95 (0.47) | 0.95 (0.56) | 0.95 (0.17) | 0.97 (0.27) | 0.95 (0.37) | 0.94 (0.44) |
| | Tana-River | 0.94 (0.32) | 0.93 (0.43) | 0.94 (0.51) | 0.94 (0.58) | 0.87 (0.2) | 0.86 (0.28) | 0.85 (0.35) | 0.85 (0.41) |
| | Turkana | 0.95 (0.24) | 0.95 (0.34) | 0.95 (0.43) | 0.95 (0.52) | 0.92 (0.14) | 0.92 (0.23) | 0.93 (0.33) | 0.94 (0.41) |
| | Wajir | 0.94 (0.37) | 0.94 (0.49) | 0.94 (0.59) | 0.95 (0.67) | 0.9 (0.22) | 0.89 (0.32) | 0.9 (0.4) | 0.9 (0.48) |
| | **Mean** | **0.94 (0.3)** | **0.94 (0.41)** | **0.94 (0.51)** | **0.94 (0.6)** | **0.91 (0.19)** | **0.91 (0.28)** | **0.9 (0.37)** | **0.9 (0.44)** |

| | | **6** | **8** | **10** | **12** | **6** | **8** | **10** | **12** |
|---|---|---|---|---|---|---|---|---|---|
| Semi-Arid Counties | Baringo | 0.95 (0.29) | 0.96 (0.42) | 0.95 (0.54) | 0.95 (0.65) | 0.95 (0.22) | 0.94 (0.36) | 0.94 (0.49) | 0.95 (0.61) |
| | Kajiado | 0.93 (0.3) | 0.93 (0.42) | 0.93 (0.53) | 0.93 (0.63) | 0.94 (0.18) | 0.93 (0.29) | 0.94 (0.4) | 0.93 (0.48) |
| | Kilifi | 0.94 (0.23) | 0.94 (0.31) | 0.95 (0.36) | 0.94 (0.41) | 0.88 (0.2) | 0.89 (0.28) | 0.89 (0.36) | 0.9 (0.42) |
| | Kitui | 0.93 (0.34) | 0.95 (0.47) | 0.94 (0.57) | 0.94 (0.64) | 0.9 (0.21) | 0.89 (0.31) | 0.88 (0.4) | 0.89 (0.47) |
| | Laikipia | 0.94 (0.24) | 0.95 (0.35) | 0.96 (0.46) | 0.96 (0.56) | 0.96 (0.17) | 0.95 (0.28) | 0.94 (0.4) | 0.93 (0.5) |
| | Makueni | 0.94 (0.34) | 0.94 (0.46) | 0.93 (0.56) | 0.94 (0.64) | 0.93 (0.22) | 0.91 (0.32) | 0.88 (0.4) | 0.89 (0.47) |
| | Meru | 0.95 (0.3) | 0.95 (0.43) | 0.95 (0.54) | 0.95 (0.62) | 0.93 (0.2) | 0.93 (0.31) | 0.92 (0.4) | 0.91 (0.47) |
| | Narok | 0.95 (0.27) | 0.95 (0.37) | 0.94 (0.45) | 0.94 (0.53) | 0.95 (0.19) | 0.95 (0.29) | 0.93 (0.39) | 0.92 (0.48) |
| | Nyeri | 0.94 (0.23) | 0.95 (0.32) | 0.96 (0.41) | 0.95 (0.49) | 0.91 (0.18) | 0.89 (0.27) | 0.88 (0.35) | 0.89 (0.43) |
| | Taita-Taveta | 0.92 (0.32) | 0.92 (0.44) | 0.92 (0.55) | 0.93 (0.63) | 0.85 (0.2) | 0.84 (0.29) | 0.84 (0.38) | 0.85 (0.44) |
| | Tharaka-Nithi | 0.94 (0.26) | 0.94 (0.37) | 0.95 (0.45) | 0.94 (0.52) | 0.92 (0.21) | 0.91 (0.3) | 0.9 (0.38) | 0.9 (0.45) |
| | West-Pokot | 0.96 (0.25) | 0.96 (0.36) | 0.95 (0.47) | 0.95 (0.56) | 0.95 (0.19) | 0.94 (0.32) | 0.93 (0.44) | 0.95 (0.54) |
| | **Mean** | **0.94 (0.28)** | **0.94 (0.39)** | **0.94 (0.49)** | **0.94 (0.57)** | **0.92 (0.2)** | **0.91 (0.3)** | **0.91 (0.4)** | **0.91 (0.48)** |

# Appendix B: Relative Importance plots for each county

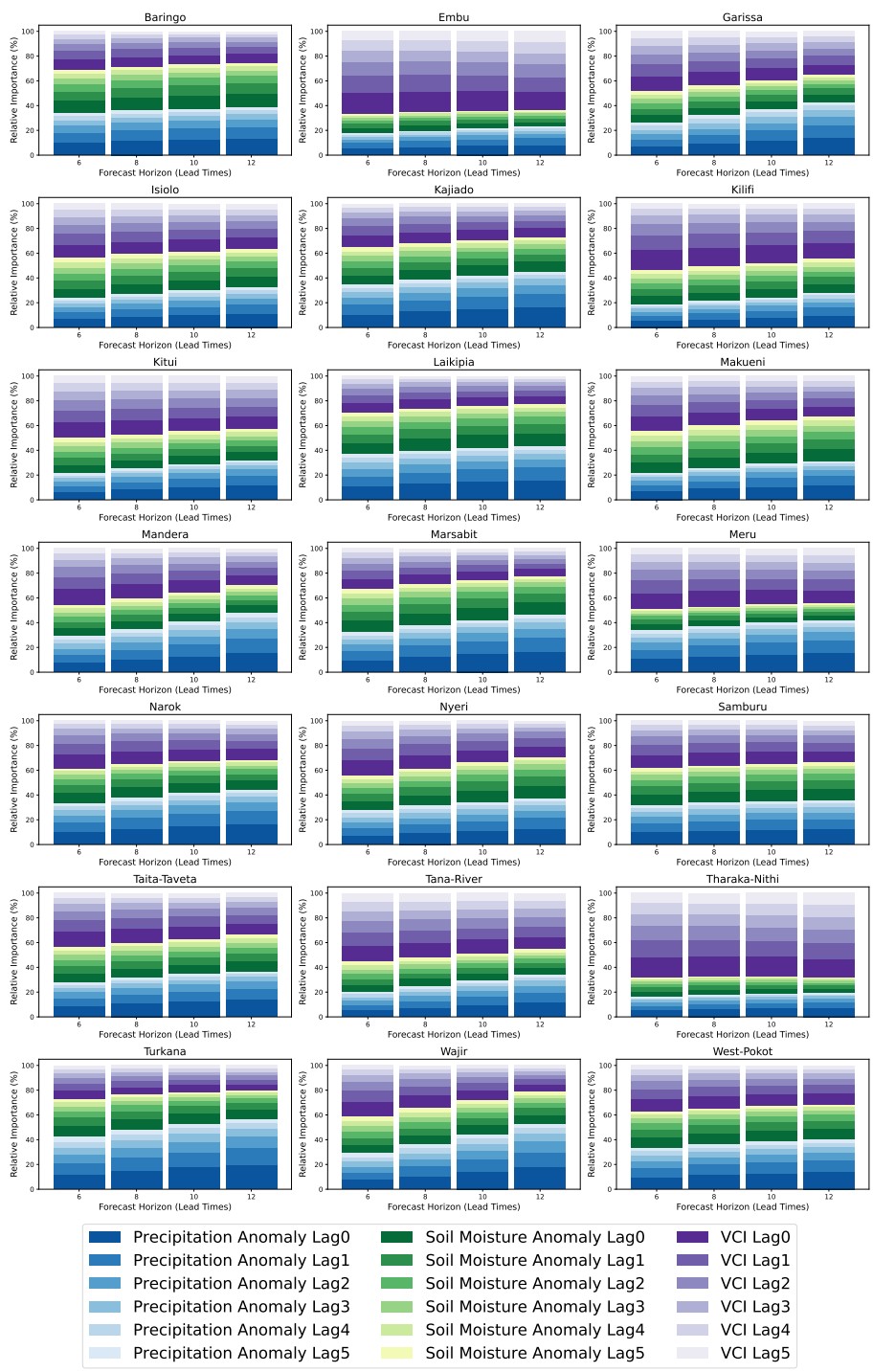

**Figure B1.** Relative Importance for each exogenous factors for each lag (0-5) variable per county. The distinct colour bands shows the overall contribution of each input variable while the varying shades within each bands shows the individual contribution each lag.

## Appendix C: Relative Importance plots for MAM and OND seasons

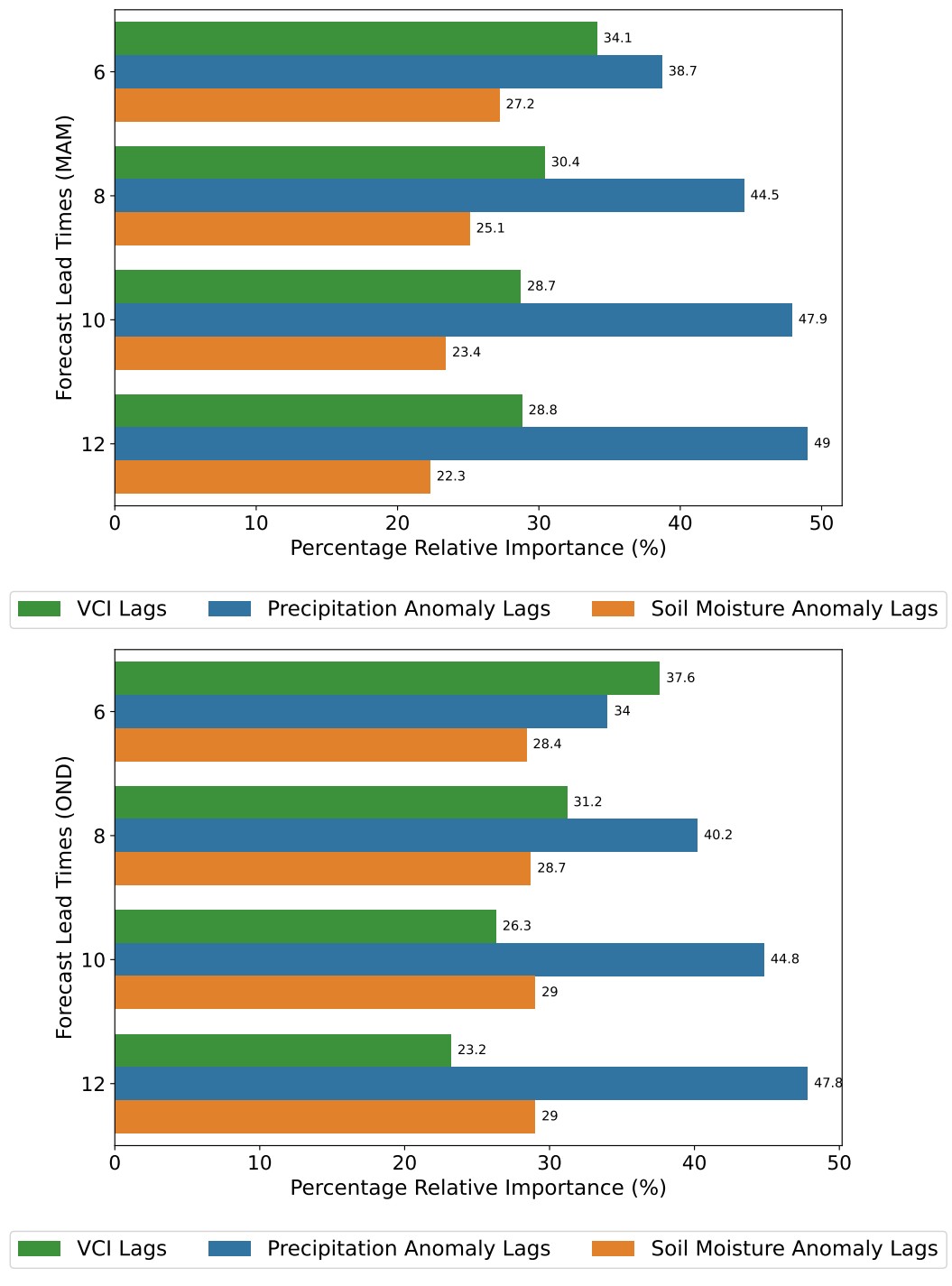

**Figure C1.** Cumulative lag relative importance plots for counties for the MAM and OND Seasons.

## Appendix D: Contour plots showing forecast performance for MAM and OND seasons

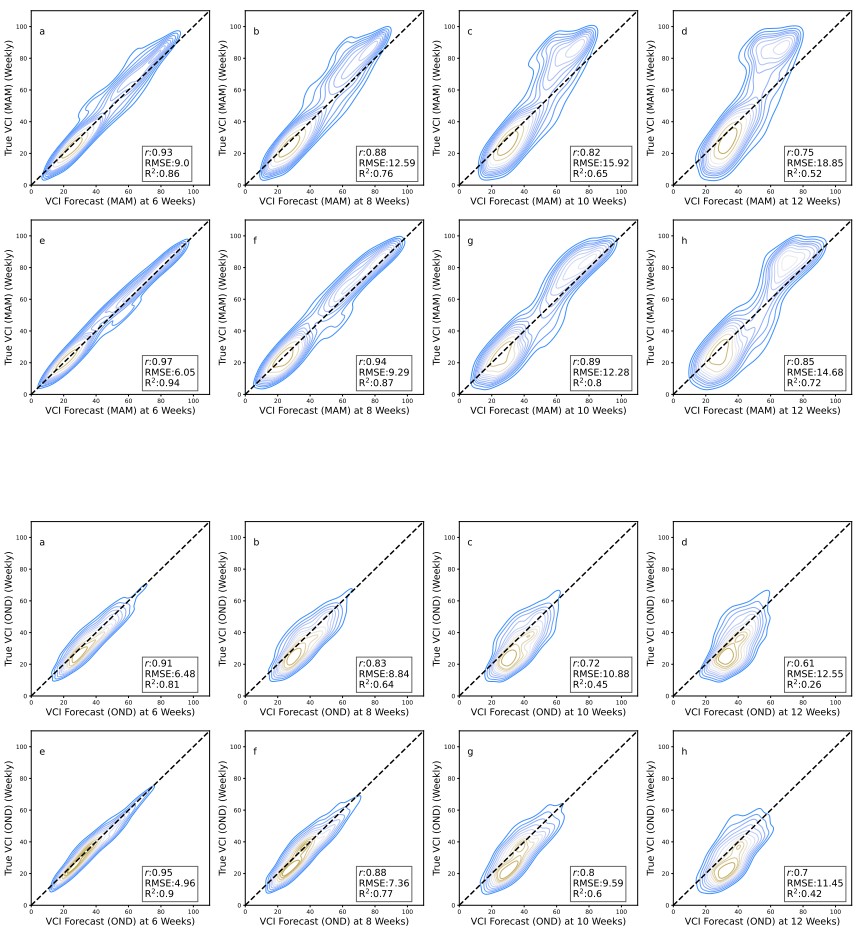

**Figure D1.** Contour plots showing VCI3M forecast against True VCI3M for MAM and OND Seasons. Plots (a,b,c,d) shows the results from the AR method with VCI3M only, (e,f,g,h) shows the overall results for BARDL modelled with lags of VCI3M plus lags of Precipitation (P3M) and Soil Moisture (S3M) Anomalies for 6, 8, 10 and 12 weeks lead time for all counties

*Author contributions.* E.E.S. lead author, data preprocessing, modelling & running BARDL method; J.M.M. data acquisition, preprocessing, cartography and feedback; A.B.B. code for AR method; A.B. code for smoothing time series data; S.O., P.R., & P.H. conceptualised the initial idea and provided supervision and feedback; The final manuscript was edited and reviewed by all authors.

*Competing interests.* All authors of the paper declare no known competing interests (financial, personal relationships) that could have influenced this study.

*Acknowledgements.* The work was funded by the UK Newton Fund's Development in Africa with Radio Astronomy (DARA) Big Data project delivered via STFC with grant number ST/R001898/1 and by the Science for Humanitarian Emergencies and Resilience (SHEAR) consortium project 'Towards Forecast-based Preparedness Action' (ForPAc, www.forpac.org), Grant Number NE/P000673/1, funded by the UK Natural Environment Research Council (NERC), the Economic and Social Research Council (ESRC), and the UK Department for International Development (DfID).

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

## List of Figures

## List of Tables