# Peer review of "Forecasting Vegetation Condition with a Bayesian Auto-regressive Distributed Lags (BARDL) Model"

_Natural Hazards and Earth System Sciences, 2021_

## Author Response (AR1)

**Forecasting Vegetation Condition with a Bayesian Auto-regressive Distributed Lags (BARDL) Model - Comments / Feedback**

**Color Code:**
- **Comments (Gray)**
- **Response (Blue)**

**Reviewer 1**

General comments

The paper presents a study aiming at forecasting vegetation conditions in arid and semi-arid environments, with up to 10 weeks lead-time in order to improve the management of droughts and anticipate their socio-economic impacts. The study uses different satellite product (NDVI, rainfall and soil moisture) to build statistical models for the forecasting of the vegetation index. The study builds on a previous paper by the same team (Barrett et al., Remote Sensing of Environment, 2020, https://doi.org/10.1016/j.rse.2020.111886 ), that used vegetation indices only and extends it using rainfall and soil moisture remote sensing data. Furthermore, the study uses a Bayesian framework for parameter estimation, that allows the determination of uncertainty on the forecast.

The paper is well written and well structured and the results are analyzed comprehensively. The conclusions appear well founded and show that the proposed methodology provides significant improvement as compared to Barrett et al. (2020), in particular in terms of lead time. Some elements in the paper could however be improved: results are compared to a benchmark model that should be better described. Some hypotheses of the work could lead to uncertainties in the results and should be discussed (use of one image in 2016 to identify grassland and shrublands, gap filling of data). Some details about methods for non-specialists of machine-learning techniques and on how figures were built are sometimes missing, precluding the good understanding of their meaning. Provided the authors address these minor comments, the paper will be suitable for publication in Natural Hazard and Earth System Sciences.

Specific comments

1/ P.2 lines 62-63: revise the sentence that is not correct.

> Response: Comment accepted; sentence has been corrected in the line 72/page 3 of the revised paper.

2/ p. 5, line 91: you use sentinel 2 data from year 2016 to identify grasslands and shrublands pixels, but are you sure that this image is representative of the whole 2001-2018 study period? It is likely that land use changes over a 18 year period, so what would be the impact of errors on the grassland and shrubland pixels on the forecasting results?

Response: Comments accepted, the issue raised has been added as a limitation in the discussion section in line 350/page 23. And sentence on the possible impact to the results have been added as well

3/ p.5 line 100-101: could you elaborate more about the gap filling method: how does it work and how the gap filling could impact the results of the forecasting model? What is the percentage of gap-filled data?

Response: Comment accepted, more details on the gap filling method have been added from line 110/page 6. But as seen in the Barrett et al. 2019 paper the gap filling method did not significantly impact forecasts.

4/ p.7 lines 122-123: could you elaborated a little more on the method used to assess the forecast probability distribution?

Response: Comment accepted and addressed from line 195/page 10 and in line 243/page 12 by adding a sentence to show how the mean forecast was calculated from the forecast distribution and used for model evaluation.

5/ p. 8 line 154: incomplete sentence?

Response: Sentence restructured in line 172, on page 9

6/ p.9 line 171-172. In order to assess the validity and robustness of a forecasting model, it is recommended to use different data for model calibration and evaluation. Could you explain more in details how you proceed with the model evaluation and if the data used for the evaluation are independent from the ones used to calibrate it.

Response: This has been made clearer, by added some details on how the split for training and test (held out) datasets was used for model training and evaluation page 10, line 195.

7/ p.9 lines 183: a figure explaining the computation of MPIW and PICP could be useful.

Response:. A diagram has been under section 3.4/page 11 to illustrate how the MPIW and PICP are computed.

8/ p.9 lines 192-194: Could the authors provide more details about the AR benchmark model. I understood that it was built using only the vegetation index, but was a Bayesian framework also used for parameter estimation? If not, could the performance of the AR model be improved is a Bayesian framework was used for parameter estimation?

Response: A description of the AR method had been added to the paper line 124/page 8. AR method used in the paper was not parameterized with Bayesian approach, however if a Bayesian method was used it might not significantly improve parameter estimation but will add

probabilistic interpretation to the AR results. The improvement in model performance was mainly attributed to additional variables. Using the Bayesian approach was to give a straightforward way to assess forecast uncertainty.

9/ p.10 lines 203-205: for non-specialists (possibly in appendix or supplementary materials) explain how the Reliability diagram and Sharpness are built.

Response: Comment accepted and details on the reliability diagram has been added from lines 227-238/page12 and 287/page18

10/ p.11 Fig. 3: explain in the figure caption how the figure is built: what are the ellipses on the figures?

Response: Comment accepted, more details on the ellipses, which are the joint distribution bins of the scatter plot between the forecasted and observed VCI3M values, have been added to the paper in line 248/page 12 and figure caption on page 13

11/ p.15 Fig 6: what is AUC in the caption and on the figures? The lines for the two models have the same colors and types. It is not easy to understand which curve is related to what? Could you also explain how this curve was built?

Response: Curves in the figure are not the same types, AR curves are dotted and the BARDL curves are solid (explained in the caption). The line colours indicate the lead times (i.e. 6, 8, 10 & 12 week ahead) which are the same for both models. The lines in figure 8 have been made thicker and more details on the AUC added in caption.

12/ p.16 Fig. 7: same remark as for Fig. 6: explain how the figures are built and what is the information they carry out.

Response: Comment accepted, details on this have been added from line 231 page 12

12/ p.16 Lines 253-256: I do not understand these explanations.

Response: Comment accepted whole paragraph has been rewritten line 285 – 297/page 18

13/ p. 17 lines 274-276: the result mentioned here was not shown before. It can be seen in the appendix but this should be mentioned.

Response: Comment accepted and well noted, and reference has been added in line 305/page 21

14/ p.18 line 280: figure ?? – please modify.

Response: Comment accepted and fixed in line 317/page 22

15/ p.17-18: a discussion of the limitations and impact of the gap filling and choice of the 2016 Sentinel 2 image to identify grassland and shrubland should be added.

Response: Comment accepted, and limitation as has been acknowledged and added to the discussion session line 350/page 23.

16/ Fig B1: Explain what is shown is these figures. They are not understandable with the current caption.
Response: Well, noted. The caption has been updated on page 26.

**Reviewer 2**

This paper proposes a new model for forecasting the vegetation condition index (VCI) based on a Bayesian autoregressive distributed lag (BARDL) model. The new model can provide the probability distribution of VCI instead of a deterministic value. In a forecasting framework, it is clear that the BARDL model can improve the current methods, as supplying a probability distribution is crucial for decision making. The BARDL model is applied to a set of counties in Kenya with arid and semi-arid conditions. VCI is forecasted from the available information about precipitation and soil moisture content, considering the current information about drought conditions. The new BARDL model is compared with the results obtained by using a deterministic AR model. The comparison is based on a set of measures that quantify both accuracy and precision. The paper offers a new method that can overcome some limitations of the current models to forecast droughts. However, the paper needs to address the comments included below before accepting it for publication.

General comments

The paper uses the vegetation condition index (VCI) to forecast droughts in Kenya. However, other indices are available like SPI, SPEI, PDSI, multi-variate standardised dry index (MSDI), the temperature condition index (TCI), the vegetation temperature condition index (VTCI), and the temperature vegetation dryness index (TVDI), among others. A discussion could be included in the paper to support the selection of VCI in the paper.

Response: The work done in this paper was in partnership with the national drought monitoring authority (NDMA)  in Kenya who are currently using VCI for monitoring drought. They have used the indicator extensively for their monthly drought reports and bulletins. In our attempt to introduce a forecast model as an additional information for bulletins, we did not want to propose a new index to them. A sentence explaining this has been added from line 50/page2

The Introduction Section focuses on three existing techniques to forecast VCI: Auto-Regression, Gaussian Processes and Artificial Neural Networks. A longer revision of the techniques used in last years to develop EWS for droughts could be included in this section, as well as other papers that develop similar tools. For example, stochastic algorithms based on different types of Markov Chains, autoregressive moving-average (ARMA), autoregressive integrated moving average (ARIMA) techniques, support vector machines, Kalman filters, multiple regression tree techniques, among others, have been used in last years to forecast droughts.

> Response: Comment well noted, the section on existing works that use similar tools have been updated from line 43/page 2

While the BARDL algorithm supplies a probability distribution, the AR model supplies a deterministic value. Therefore, the comparison between the two models is not straightforward. In the paper, a confidence interval for the AR model is estimated from RMSE and z-score. However, this is a simplified way to estimate the prediction uncertainty, supplying a constant confidence interval regardless the magnitude of both VCI and the explanatory variables. This step is very important to compare BARDL results with AR results in a proper way. In addition, the methodology to compare both models should be clarified in the paper, as it is not clear how most of measures used to quantify accuracy and precision have been applied to the probabilistic forecast supplied by BARDL.

> Response: This is well noted comment has been accepted and has been addressed by added a sentence to explain how the mean forecast were computed from forecast distribution before used for model evaluation line197/page10 and line 244/page 12

The Discussion Section should be rewritten, as in its current form it is mostly a mixture of conclusions with some additional results considering seasonality.

The Conclusions Section could be extended to summarise the main findings of the study.

> Response: Comment accepted and well noted, the discussion the discussion and conclusion have be restructured pages 21 - 23

Specific comments:

Abstract: Some sentences could be included in the abstract about the case study used in the paper.

> Response: Comment accepted and fixed by adding the sentence in line 10/page 1

14: The acronym AR has not been introduced in the paper at this point yet.

> Response: Comment noted has been fixed by expanding the AR and making sure its repeated subsequently.

30: The acronym USAID is not introduced in the paper and could be explained at this point.

Response: Comments noted USIAD has been expanded in the footnote on page 2

46: The ARDA model has been applied to assess droughts previously, such as Zhu et al. (2018). References to previous studies in which the ARDA technique is applied to droughts should be included in the paper.

Response: Well noted will be considered, however paper focused on Hydrological Droughts in river basins and not vegetation conditions.

51: The paper proposes the use of a Bayesian framework in the ARDA model to incorporate the prior knowledge about model parameters in the analysis, obtaining a probability distribution for VCI results.  Bayesian networks have been also applied to develop a long-term drought forecast (Shin et al., 2019), supplying probabilistic results that can assess forecast uncertainties. A discussion could be included in the paper, stating the benefits of a BARDL model compared to Bayesian networks.

Response: Comment noted and will be considered however, the results in this paper are also for Hydrological drought and not comparable to agricultural drought indicators.

Section 2.1: Some information about the number of counties considered in the study could be included in this section, as well as the number of counties that are arid and semi-arid. In addition, some information about the area in km2 that is considered in the study could be useful for the reader.

Response: Comment accepted and well noted, the more details has been added in section 2.1 line 66/page3 and in the caption

70: 'estimates' should be changed to 'estimate.

Response: Comment noted and fixed in line79/page5

98-99: The description of NVIi and NDVIi variables should be included in this paragraph too.

Response: Comment noted and  fixed in line 108/page 6

103-104: 'long term' should be changed to 'long-term'.

Response: Comment noted and fixed in line 116/page 6

111: The acronym AR has been introduced in the paper above.

Response: Comment noted and fixed in line 130/page8

118: A discussion could be included about the selection of the OLS method for estimating parameters of ARDL. Some other methods are also available.

Response: Comment not too clear because OLS was not used for the ARDL in this paper.

131 – Eq. 3: The variable subscripts should be revised in Eq. 3. Dt-q seems to be the drought indicator in a constant time step t-q, which seems to be constant in the first summation regardless the value of i. Similarly, Pt-p and St-p seem to be constant values in the summations. In addition, the regression coefficients are also constant values in the summation, though they could change in terms of i. A discussion should be included about the use of constant values in summations.

Response: Comment noted, and the equation has been redone make clear line150/page8

137 – Eq. 4: How does Xt-i represent several variables? How can i vary from 0 to i?

Response: Comment noted, these subscripts represent lagged order of the input variables, but it will be amended to make clearer in the paper. Fixed from line151/page8

143-146: The variable theta should be explained to readers in this paragraph.

Response: Comment noted and fixed in line 162/page9

145-146: The term P(Xt) is ignored because it is difficult to compute. This is not a proper statement for ignoring a variable in a research paper.

Response: Comment noted the statement has been restructured in line 164/page9

152-154: An analysis should be done to fix the distribution function that best characterises the regression parameters. Why mu is set to 0 and sigma to 0.5?

Response: This parameter was selected after the model optimisation process was done. This was because these parameters gave best forecast results and with minimum error.

153-154: Something is missing in this sentence.

Response: Comment noted sentence has been rewritten in line 173/page9

164: This is not the standard form of AIC.

Response: Comment noted and has been fixed in line 187/page10

161-163: Some figures could be included in the paper to show how a time lag of 6 weeks obtains the best AIC and R2 results.

Response: Comment noted, and I acknowledge the importance of this, however accessing my former university's server to access the code to re-run and generate figures has been a little challenging and may delay resubmission.

168: What is i? What is y hat?

Response: Comment noted these have been explained in line 191/page 10

176: The R2 measure of Eq. 9 is not a good measure to quantify accuracy of forecasts.

Response: We considered the R2 score for the work because in addition to the knowing deviation of the forested values from observed values (RMSE) we also needed to test the goodness of fit or the variation in dependent variables captured or explained by model.

188-189: What is m?

Response: M is also refers to number of sample, but it has been changed to N 211/page 10

196: 'inputs' should be changed to 'input'.

Response: Comment noted and fixed in line 220

213: How r, R2 and RMSE are calculated for the BARDL model? The BARDL model supplies a probability distribution, but observations are deterministic.

Response: To determine these metrics the mean of the forecast probability distributions were used. Address in lines 197/page 10 and  244/page12.

214: R2 is not a good measure of forecast accuracy. RMSE is more adequate than R2. Therefore, the gain in performance metrics could be assessed with RMSE. However, the BARDL model supplies a probability distribution of VCI. How do you obtain a RMSE value from the comparison between probability distributions and deterministic values of observations?

Response: We used the R2 because we needed to test the goodness of fit and the variation in dependent variables captured or explained by our model. The R2 and RMSE values were determined with the means (Average) of the forecast probability distribution.

Figure 3: What do the coloured lines mean?

Response: The contour lines represent the density and bins of the joint distribution plot. An explanation has been added in line 248/page12.

222-224: The R2 values do not correspond with the values shown in Table 2.

Response: The R2 values do not correspond because the table 2 is showing R2 values for the separate Arid and Semi-Arid zones and not the overall as seen in the figure.

Table 2: This table could be summarised in a figure.

Response: Comment noted, and will be considered. This nature of the data in the table will make the plot difficult to interpret visually.

229-231: The table in Appendix A could be summarised in a figure and included in the main text of the paper, in order to analyse the comparison between the two models. The results included in Table 1 show that PICP values are smaller for AR than for BARDL, meaning that a greater number of observations are out of the confidence intervals for BARDL. This result should be discussed in the paper. In addition, most of PCIP values for the BARDL model are smaller than 94-96 %, in contrast to the statement of line 229.

Response: Comment noted, comment and sentences on this have been added in lines 267/age 15-16 and further discussed from line 332 – 335/page 22

Figure 5: Please use the same y-axis scale in each row to compare the AR and BARDL results. The dashed line of the left column differs from the dashed line of the right column, though observations do not change. The green line represents the forecast. What is such a forecast for the BARDL model given that it supplies a probability distribution?

Response: Comment well noted, the y-axis have been set to the same scale in figure 7/page16. The difference in the dashed line is due to shift in time series data when creating observed led time datasets.

235: A drought is forecasted when VCI3M values are smaller than 35. This is straightforward for the AR model, as it is deterministic. However, how do you apply this criterion to the BARDA outputs considering probability distributions?

Response: Comment noted, the criterion was applied to the mean of the forecast distribution from the bayesian model.
We have noted that details on the use of the mean forecast distribution model evaluation are missing the narrative and will be addressed. line 247/page12

251-253: The BARDL lines lie above the main diagonal of the reliability diagram. This means that the probabilities supplied by the BARDL model tend to underestimate droughts. A comment about this point should be included in the paper.

Response: Comment noted, this was an omission in the paper and the details on this have now been outlined in the paper from line 290-294/page18

253-256: The sharpness diagrams are mostly flat for 10 and 12 weeks. The low values close to 1 means that the BARDL model is not able to forecast droughts. Therefore, the BARDL model is useful to forecast droughts with 6 weeks ahead, but it is not for 10 and 12 weeks. A comment about this point should be included in the paper.

Response: Comment well noted, the details on this point have been outlined in line 293/page 19.

Figure 7: The sharpness diagram should plot percentages in the y axis.

Response: Comment well noted, and has been fixed in figures 9 and 10/pages 19 and 20

266-276: These two paragraphs could be moved to the Conclusions Section.

Response: Comment well noted, this has been done see page 23 from line 355 - 362

273-274: The Authors state that the BARDL model gains 2 weeks based on the results of R2. However, more measures should be taken into account to conclude such a statement.

Response: Comment well noted, these was demonstrated in the figure 4 where r2 score seen in 6 week forecast of AR were equivalent or close to r2 scores seen in 8weeks for BARDL, this can also been seen in table 2. And also mentioned in line 311/page 21

275-276: This statement is not clear from the results included in Section 4.

Response: Reference has been made to the figures 11, B1 and C1 and discussed in lines 318-320/page22

280: The number of the figure is missing.

Response: Comment noted and fixed in line 321/page 22

284-295: This paragraph with figures of Appendixes C to F could be extended to form a new section 4.6 devoted to the seasonality analysis.

Response: The figures and paragraphs have been moved to Results section in line 260-261/page 14  and lines 298-299/page 19

---

## Author Response (AR2)

Dear All,

Thank you all for the review and feedback they were all well appreciated. Please see my point-by-point response to all the comments below.

**Reviewer 1**,

Line 11-12: should be: The objective of this research was to build on this work by developing an improved model that forecasts vegetation conditions at longer lead times

**Response:** The sentence has been corrected as *"The objective of this research was to build on this work by developing an improved model that forecasts vegetation conditions at longer lead times."* to reflect the comment. Page 1

Line 157; should be "Equation (4) can be re-written as"

**Response:** The typo has been fixed in line 161 on page 9

Line 173: thus, the need to use -> verb missing

**Response:** This was fixed to read as *"Computing equation 7 requires very complex integrals (Lambert, 2018) thus the HMC sampling algorithm (Hoffman and Gelman, 2014) was used for estimating the model parameters"* in line176 on page 9

**Reviewer 2**,

The authors have addressed most of the comments satisfactorily. However, a set of minor suggestions are still needed to consider the paper for its publication.

The selection of VCI for monitoring droughts is based on the previous experience of the national drought monitoring authority (NDMA) in Kenya. However, a discussion about the strengths and drawbacks of VCI compared with other indices should be included.

**Response:** The comment was addressed by including a sentence in the introduction section on the strengths of VCI.  A reference  to a paper  (See below) on a comparative analysis between VCI and other Drought indicators done by our team soon after this paper was submitted has also been added. Line 50 -57 on pages 2 - 3

*Bowell, A., Salakpi, E. E., Guigma, K., Muthoka, J. M., Mwangi, J., and Rowhani, P.: Validating commonly used drought indicators in Kenya, Environmental Research Letters, 16, 084 066, https://doi.org/10.1088/1748-9326/ac16a2, 2021.*

In the case study, information about its extent in km2 is still missing.

Response: The comment was addressed by adding the sentence "*The ASAL regions make up about 80% (46,000km2) of Kenya's total land area (Marigi et al., 2016)*" on the estimated area covered by ASAL region in Kenya under the section on study area on page 3

51. Please change 'it's' to 'it is'.

**Response:** The sentence has been fixed

108. The description of VCIi is still missing.

**Response:** The description of VCIi has been added to eq. 2 on page 6

Eqs. 4 and 5. No changes are appreciated between the first and revised submissions of the paper.

**Response:** Eq 4 was initially Eq 3, and the subscripts *I,* which were used to indicate the lags, were changed to p and q to reflect the ARDL(p,q) process. Eq 5 is a simplified version of eq 4 to show how the ARDL(p,q) fits in the Bayesian context.

Figure 4. A legend is missing to show the value of the contour colours.

**Response:** The colour of the contour lines indicated the level of correlation between predicted and observed values. The sentence has been added to explain this in the caption of Figure 4 on page 13

255-260. The R2 values do not correspond with the values shown in Table 2. A comment should be added to the text to explain to the readers the source of the differences.

**Response:** A comment explaining the difference has been added to the description of the table on page 16

Figure 7. The paper (figure caption or text) should include that the forecast line (green line) represents the mean value of the BARD model outputs.

**Response:** The caption text for this figure has been amended on page 17

332-333. Please, improve the understandability of this sentence.

**Response:** The sentence has been restructured as " *The sharpness plots in figure 9 also indicated that the model was generally sensitive to identifying drought events more frequently till eight weeks ahead.*" to make it clear to the reader. lines 335-338 on page 23